# Profiling the *Escherichia coli* membrane protein interactome captured in Peptidisc libraries

Michael Luke Carlson[1†], R Greg Stacey[2†], John William Young[1†], Irvinder Singh Wason[1], Zhiyu Zhao[1], David G Rattray[2], Nichollas Scott[2], Craig H Kerr[2], Mohan Babu[3], Leonard J Foster[2], Franck Duong Van Hoa[1*]

[1]Life Sciences Institute, Department of Biochemistry and Molecular Biology, Faculty of Medicine, University of British Columbia, Vancouver, Canada; [2]Michael Smith Laboratory, Department of Biochemistry and Molecular Biology, Faculty of Medicine, University of British Columbia, Vancouver, Canada; [3]Department of Biochemistry, Faculty of Science, University of Regina, Regina, Canada

**Abstract** Protein-correlation-profiling (PCP), in combination with quantitative proteomics, has emerged as a high-throughput method for the rapid identification of dynamic protein complexes in native conditions. While PCP has been successfully applied to soluble proteomes, characterization of the membrane interactome has lagged, partly due to the necessary use of detergents to maintain protein solubility. Here, we apply the peptidisc, a 'one-size fits all' membrane mimetic, for the capture of the *Escherichia coli* cell envelope proteome and its high-resolution fractionation in the absence of detergent. Analysis of the SILAC-labeled peptidisc library via PCP allows generation of over 4900 possible binary interactions out of >700,000 random associations. Using well-characterized membrane protein systems such as the SecY translocon, the Bam complex and the MetNI transporter, we demonstrate that our dataset is a useful resource for identifying transient and surprisingly novel protein interactions. For example, we discover a trans-periplasmic supercomplex comprising subunits of the Bam and Sec machineries, including membrane-bound chaperones YfgM and PpiD. We identify RcsF and OmpA as bone fide interactors of BamA, and we show that MetQ association with the ABC transporter MetNI depends on its N-terminal lipid anchor. We also discover NlpA as a novel interactor of MetNI complex. Most of these interactions are largely undetected by standard detergent-based purification. Together, the peptidisc workflow applied to the proteomic field is emerging as a promising novel approach to characterize membrane protein interactions under native expression conditions and without genetic manipulation.

DOI: https://doi.org/10.7554/eLife.46615.001

*For correspondence: fduong@mail.ubc.ca

†These authors contributed equally to this work

## Introduction

Proteins control biological systems in a cell. While many perform their functions independently, the majority of proteins interact with others to achieve their full biological activity. Characterizing protein-protein interaction networks (the interactome) has traditionally been accomplished by methods such as affinity purification coupled to identification by mass spectrometry (AP/MS) (*Arifuzzaman et al., 2006*; *Hu et al., 2009*; *Babu et al., 2012*; *Babu et al., 2018*), protein fragment complementation assays (*Rochette et al., 2015*; *Tarassov et al., 2008*), or yeast two-hybrid screening (Y2H) (*Rajagopala et al., 2014*). While high-throughput, these methods are quite often limited in their scope by poor scalability because bait proteins must be independently tagged. The addition of these tags can also have uncontrolled effects on proteins such as disrupting binding sites, altering

localization, stability, and thereby the accurate prediction of the interactome. Co-fractionation methods, such as protein-correlation-profiling (PCP) in combination with quantitative proteomics methods, such as label-free quantitation (LFQ) or stable isotope labeling of amino acids in cell culture (SILAC), are therefore emerging as an attractive alternative to identify protein complexes under native expression conditions and without genetic manipulation (*Kristensen et al., 2012*; *Scott et al., 2017*; *Havugimana et al., 2012*). Fractionation of a proteome under these native conditions, followed by quantitative proteomic analysis of co-fractionation profiles, allows identification of protein complexes through a principle of 'guilt-by-association'. This method can generate thousands of potential interactions in a single experiment, and incorporation of SILAC multiplexing allows simultaneous comparison of multiple states of the interactome (*Kristensen et al., 2012*; *Scott et al., 2017*).

While co-fractionation has been successfully applied to soluble proteomes, characterization of the membrane proteome has lagged. This is largely due to the hydrophobic nature of membrane proteins and their sequestration in the lipid membrane. To extract this water-insoluble proteome, it is necessary to solubilize the lipid bilayer with the aid of detergents or amphipathic co-polymers such as styrene maleic acid (SMA) (*Dörr et al., 2014*). When mild detergents are employed, membrane protein complexes can be directly detected following their separation by techniques such as size exclusion chromatography, density gradient centrifugation (*McBride et al., 2017*) or blue-native gel electrophoresis (*Scott et al., 2017*; *Heide et al., 2012*). However, even the mildest detergents tend to decrease protein stability while increasing protein aggregation (*Yang et al., 2014*). In fact, prolonged exposure to those detergents tends to delipidate proteins and alter their conformation, which can have confounding effects on membrane protein complex stability. As an additional drawback, micelles of detergent must be removed from all samples before analysis by mass spectrometry, which often decreases protein identification (*Yeung and Stanley, 2010*; *Bechara et al., 2015*; *Bao et al., 2013*; *Yang et al., 2014*). Thus, while a great deal of useful data has been generated using detergent-based proteomics analysis, there is still a pressing need for novel methods that are unencumbered by detergent side-effects.

We recently developed the peptidisc as a novel membrane mimetic scaffold to keep membrane proteins water-soluble (*Carlson et al., 2018*). The peptidisc is formed when multiple copies of the 4.5 kDa amphipathic scaffold $NSP_r$ (also called Peptidisc peptide) wrap around the solubilized membrane proteins. Reconstitution occurs spontaneously upon removal of detergent, incorporating both endogenous lipids and solubilized membrane proteins into detergent-free particles. The number of scaffolds adapts to fit the diameter of the protein target without bias toward large protein complexes. The end result is peptidisc particles that are stable, free of detergent effects, and soluble in aqueous solution (*Carlson et al., 2018*). Our previous work has shown that the peptidisc is able to stabilize both inner and outer membrane proteins of *Escherichia coli*.

In this study, we apply the peptidisc to the trapping of the bacterial cell envelope proteome into water-soluble particles. This is performed by reconstituting the heterogeneous membrane protein mixture immediately after its extraction from the cell envelope with mild detergent. This process minimizes protein dissociation and denaturation because it limits exposure to detergent and thereby protein delipidation. The membrane proteome trapped in the peptidisc library is water-soluble and stable during prolonged incubations. This library is then fractionated by high-resolution size exclusion chromatography (SEC) in the total absence of detergent. Application of the PCP workflow, which includes stable isotope labeled amino acids in cell culture (SILAC) and mass spectrometry (LC/MS-MS), allows us to precisely characterize the content of the peptidisc library across the various fractions. When the peptidisc library from the raw *E. coli* cell envelope is analyzed this way, we identify and quantify 1209 unique proteins, of which 591 are predicted to be directly membrane integrated. From these 1209 proteins, we predict 4911 binary interactions - each characterized by a degree of precision. Our interaction list is hereafter called the peptidisc interactome.

To computationally validate the precision of the peptidisc interactome, we benchmark the dataset against the recently published *E. coli* cell envelope interactome ('CE') (*Babu et al., 2018*) and two other unpublished interactomes collected for that earlier study ('validating interactomes'). We also measure the biological plausibility of the peptidisc interactome by determining enrichment for shared gene ontology terms, binding domains, and correlation of growth phenotype (*Erickson et al., 2017*; *Mosca et al., 2014*). We also compare the peptidisc-reconstituted membrane proteome against a membrane proteome prepared using the SMA polymer instead of detergent.

We find, however, that large membrane protein complexes are better preserved in the peptidisc workflow.

Guided by the peptidisc interactome datalist, we select three well-characterized membrane protein complexes in order to discover novel interactions. With the Sec translocon, we validate association of SecY with the membrane chaperones YfgM and PpiD. This interaction can be isolated in detergent but only when all subunits are simultaneously over-produced. Remarkably, we also discover significant correlation between certain subunits of Sec and Bam complexes, suggesting an astonishing network of protein associations spanning across the entire bacterial cell envelope. We confirm this observation using SILAC AP/MS, thereby providing direct evidence for the Bam-Sec super-complex. Continuing with the Bam complex, we show that all five subunits are captured in peptidisc in addition to two other interactors - RcsF and OmpA. These interactions were previously inferred from genetic and indirect cross-linking experiments, but direct association was not formally demonstrated (*Hart et al., 2019*; *Konovalova et al., 2014*). Accordingly, these interactions are much less apparent in detergent. Finally, with the ABC transporter MetNI, we find that binding of the substrate binding protein MetQ depends on its N-terminal lipidation The importance of the MetQ lipid anchor is novel and is unique case among Type I ABC transporters. Moreover, we identify NlpA, also called lipoprotein 28, as a bona fide novel interactor of the MetNI complex.

Altogether, this work validates the peptidisc library workflow as an efficient method for capturing and stabilizing the membrane proteome into soluble particles. The method enables high-throughput detection of detergent-sensitive membrane protein interactions. When combined with rigorous experimental validation, the peptidisc interactome is revealing novel and transient interactions, many of them of fundamental importance to the transport process and biogenesis mechanism of the cell envelope.

## Results

### Capture of the *E. coli* membrane proteome in peptidisc

The peptidisc-based SEC-PCP-SILAC workflow is presented in *Figure 1*. To identify the optimal solubilization of the *E. coli* cell envelope, we employed three different non-ionic detergents (DDM, LDAO and β-OG) and one ionic detergent (DOC). Each was tested using *E. coli* crude membranes enriched for an inner membrane protein marker, MsbA. Upon removal of aggregate by ultracentrifugation, the solubilized membrane proteomes were incubated with Peptidisc peptide at the ratio 2:1 (g/g). Formation of peptidisc libraries was initiated by detergent dilution followed by filtration and concentration on a spin column with a cut-off of 100 kDa. The overall content of each library was then compared to the original detergent extract using SDS-PAGE and Coomassie blue staining (*Figure 2A*). Visual analysis indicated that most, if not all, of the proteins solubilized in detergent were present in the peptidisc library (*Figure 2A*). However, the best extraction of the marker protein MsbA was seen with DDM; therefore, this detergent was employed in the subsequent extraction studies.

We next assessed the amenability of our peptidisc library preparations to fractionation by size-exclusion chromatography (SEC). We compared the SEC fractionation profiles between the *E. coli* membrane proteome solubilized in DDM versus the *E. coli* membrane proteome trapped in peptidisc library (*Figure 2B* and *Figure 2C*). The overall protein profiles of each fractionation were comparable, as assessed by SDS-PAGE. The richest protein fraction from each (i.e. fraction #12; *Figure 2B* and *Figure 2C*) was analyzed by mass spectrometry. A total of 125 proteins and 162 proteins were identified from the detergent and peptidisc samples with ~85% overlap between the two, respectively (*Figure 2D*).

To verify that individual membrane proteins and complexes were trapped in discrete peptidisc particles - rather than being non-specifically clustered together - we isolated MsbA from the peptidisc library via a Ni-NTA pulldown. Analysis by SDS-PAGE revealed that his-tagged MsbA is efficiently isolated from the peptidisc library preparation (*Figure 2F*). Native PAGE analysis revealed that the purified MsbA is, as expected, homogenous (*Figure 2G*). These results strongly indicate that individual membrane proteins are efficiently trapped in discrete peptidiscs.

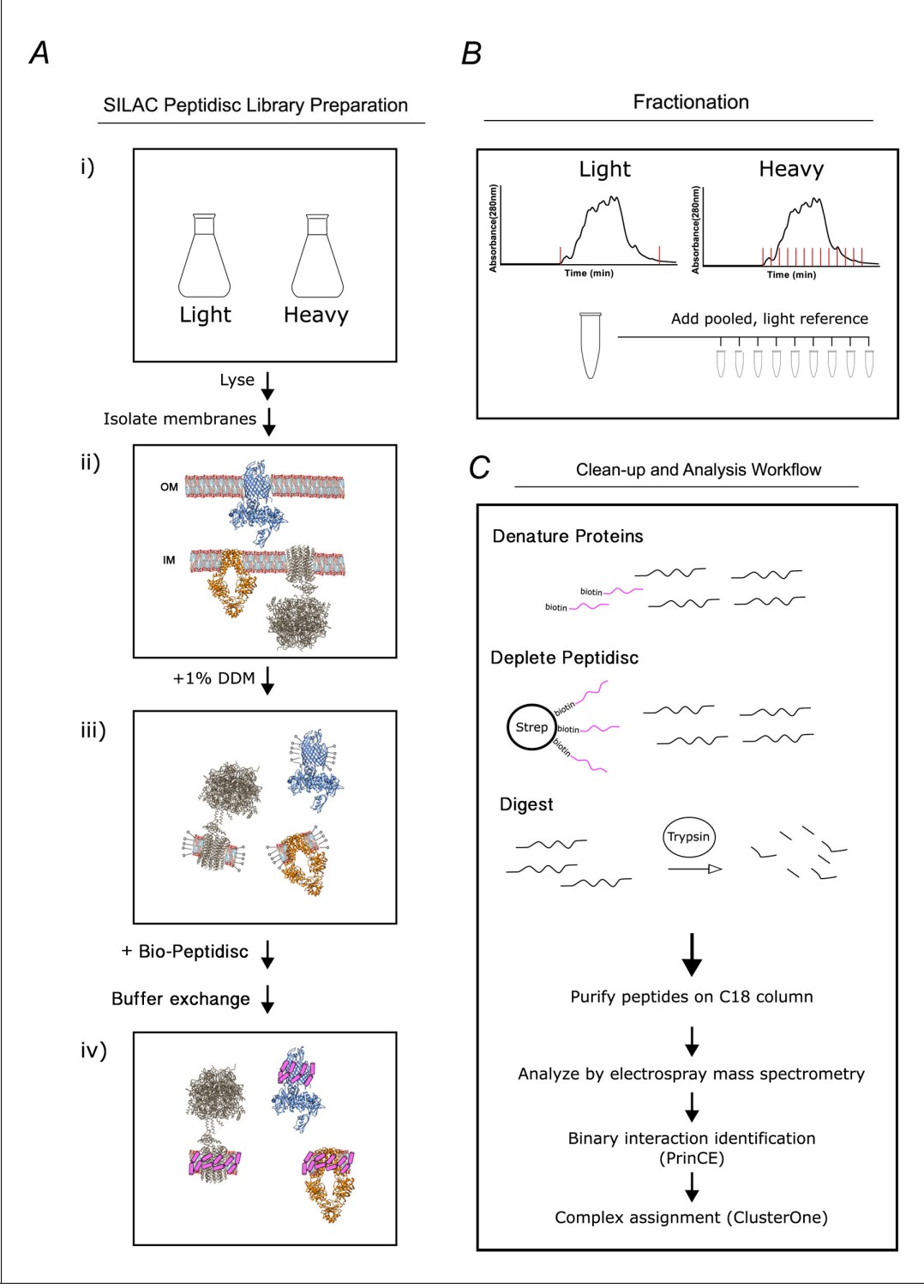

**Figure 1.** Overview of the peptidisc-based SEC-PCP-SILAC workflow. (**A**) Identical *E. coli* cultures are labeled in SILAC media (i), lysed with french press and crude membrane fraction isolated by ultracentrifugation (ii). Membranes are solubilized in non-ionic detergent (DDM) (iii), transferred into biotinylated Peptidisc peptides (Bio-Peptidisc) solution, and then filtered to remove excess peptide and detergent (iv). (**B**) The light and heavy peptidisc libraries are separated by high-resolution SEC in detergent-free buffer. The light fractions are pooled and aliquoted into the heavy fractions as an internal quantification standard. (**C**) Proteins in each fraction are denatured, depleted for Bio-Peptidisc peptides, digested, and analyzed by LC-MS/MS. Maxquant is used to identify peptides and to quantify heavy protein enrichment in each fraction. Binary protein interactions are identified from the co-
*Figure 1 continued on next page*

*Figure 1 continued*
elution data using the prediction of interactomes bioinformatics pipeline (PrinCE). Binary interactions are subsequently segregated into predicted complexes using the ClusterONE algorithm.
DOI: https://doi.org/10.7554/eLife.46615.002

## Fractionation of the SILAC-labeled peptidisc library

We next prepared light and heavy isotopically labeled membrane proteomes and reconstituted them into peptidisc libraries. Both libraries were separated by high-resolution size exclusion chromatography using two silica-based BioSep4000 columns placed in tandem. The light fractions were pooled and an equivalent volume was added to each heavy fraction before trypsin digestion and electrospray mass spectrometry analysis. For each fraction, proteolytic peptides were identified and SILAC ratios were determined using Maxquant. Two biological replicates were performed, resulting in the identification of 1209 proteins across the 54 fractions (raw data presented in *Supplementary file 1*). As expected, a large fraction of these proteins (591 proteins) are predicted to be associated with the cell envelope (*Table 1*). In addition to predicted cell envelope proteins, there were also soluble proteins which are known to associate into macromolecular complexes, such as the ribosome or GroEL complex. Previous reports have shown that these large assemblies are prone to co-sedimenting with the cell membrane fraction during the ultracentrifugation step after cell lysis, thus explaining their presence in our peptidisc library preparations (*Papanastasiou et al., 2013*; *Papanastasiou et al., 2016*).

To compare the peptidisc library with another detergent-free fractionation method, we solubilized the same *E. coli* membrane with the styrene maleic acid co-polymer (SMA). The SMA polymer directly solubilizes membrane lipids and captures proteins into styrene maleic acid lipid-protein nanoparticles (SMALPs)(*Dörr et al., 2014*). Following SEC fractionation of the SMALPs library and MS analysis, we identified 1576 proteins across 54 fractions (raw data presented in *Supplementary file 2*). There was good reproducibility between replicates and a large part of the identified proteins (705 proteins) was predicted to be associated with the cell envelope. The overlap in protein content between the SMALPs and peptidisc library was excellent, with 1026 proteins shared between the two libraries. Furthermore, the overall distribution of proteins in each library, as classified according to their gene ontology annotations and their originating compartment, was nearly identical (*Figure 3A* and *Figure 3B*). Thus, SMALPs and peptidiscs are both suitable for solubilization and detergent-free fractionation of the *E. coli* cell membrane. Importantly, the similar repertoire of proteins identified in both the SMALPs and peptidiscs libraries indicate that detergent solubilization followed by immediate reconstitution into peptidisc results in comparable capture efficiency of the membrane proteome as direct solubilization in the SMA polymer.

## Large membrane protein complexes are captured in the peptidisc library

To determine if the peptidisc is able to capture membrane protein complexes, we compared the co-fractionation profiles of three well-characterized protein assemblies after encapsulating the membrane library in either SMALPs or peptidiscs (*Figure 3C*, *Figure 3D* and *Figure 3—figure supplement 1*). In all three cases, the complexes appeared more stable in the peptidisc library than in SMALPs. In the SMALPs library, the ATP synthase complex was dissociated, causing its protein subunits to elute separately (*Figure 3C*). In contrast, the ATP synthase was preserved in peptidiscs (*Figure 3D*). The Bam complex (BamABCDE) was solubilized with the SMA polymer, but here the elution profiles for the individual subunits showed weak correlation, suggesting at least partial dissociation of the complex (*Figure 3—figure supplement 1*). In contrast, each Bam subunit presented an almost identical co-fractionation pattern in peptidisc (*Figure 3—figure supplement 1B*). Similarly, the individual subunits of the respiratory chain complex (NuoABCDGI) showed a higher degree of correlation in their fractionation profiles in peptidisc versus SMALPs (*Figure 3—figure supplement 1D and C*). These results indicate that while SMA is an effective solubilization agent, it does not stabilize large membrane protein assemblies. The peptidisc is therefore better suited for stabilization of large, multiprotein complexes.

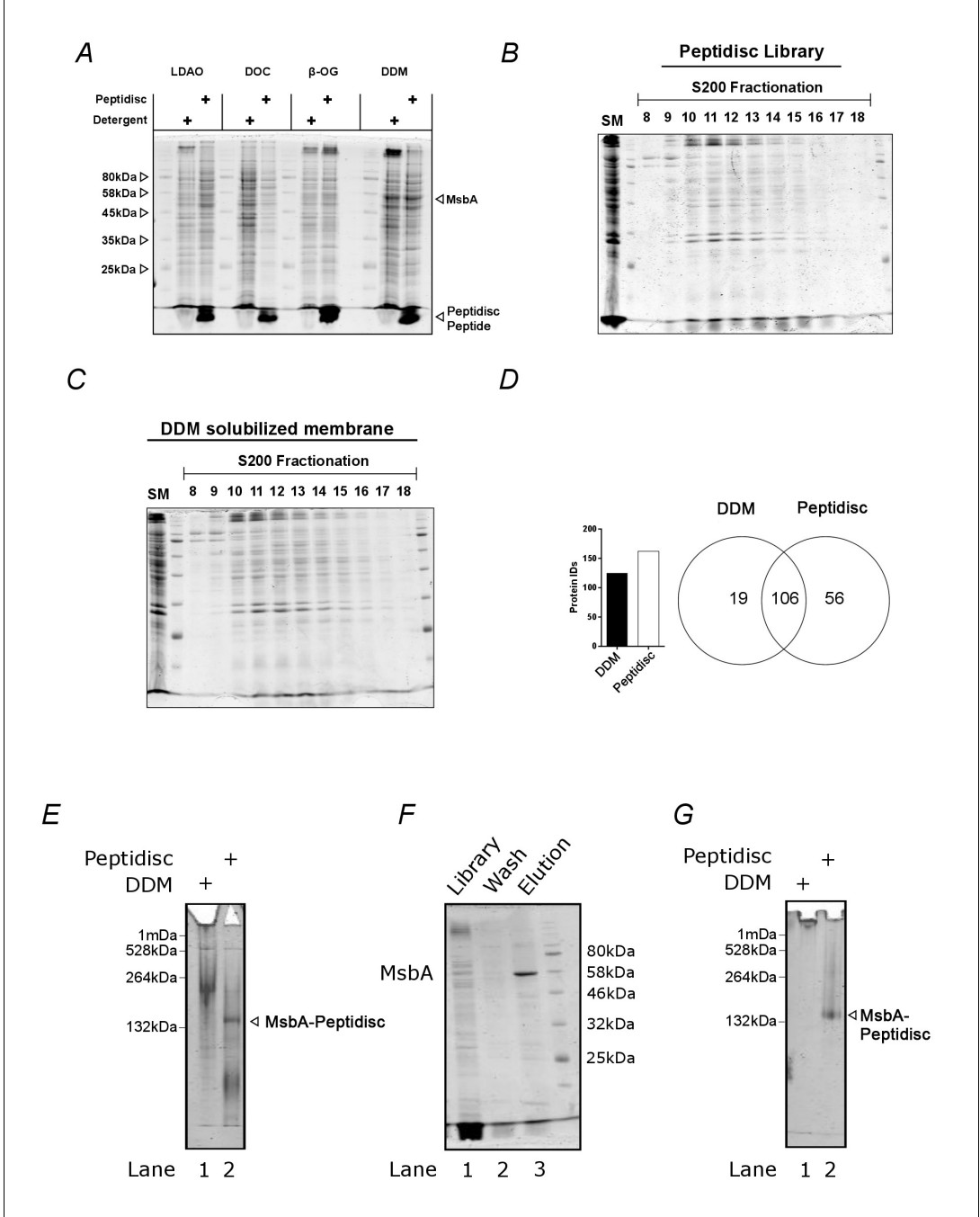

**Figure 2.** The peptidisc captures detergent solubilized membrane proteins with high efficiency. (A) SDS-PAGE analysis of detergent solubilized *E. coli* crude membrane before and after reconstitution into peptidiscs. The crude membrane preparation was solubilized in either 1% n-dodecyl-beta-maltoside (DDM), 3% ß-octyl glucoside (ß-OG), 1% sodium deoxycholate (DOC), or 1% lauryldimethylamine-N-oxide (LDAO), followed by reconstitution into peptidiscs by dilution and buffer exchange. (B) Protein number and overlap after SEC-fractionation of DDM extract and peptidiscs library prepared from DDM extract. A total of 20 fractions were collected, and the fraction containing the highest concentration of protein (fraction 12) analyzed by electrospray mass spectrometry in triplicate. The mass spectrometry data was searched together in MaxQuant. (C) SDS-PAGE analysis of native *E. coli* membranes incorporated into peptidisc after fractionation by size exclusion chromatography in detergent-free buffer. (D) As in C, with membranes solubilized in DDM and fractionated in buffer supplemented with DDM. (E) Clear native (CN)-PAGE analysis of crude membrane solubilized in DDM (Lane 1) or in peptidiscs (Lane 2). (F) The peptidisc library containing overexpressed MsbA (Lane 1) was bound to Ni-NTA beads, washed in Buffer A (Lane 2), and eluted in Buffer A + 250 mM imidazole (Lane 3). Samples were analysed by SDS-PAGE. (G) CN-PAGE analysis of MsbA purified in DDM (Lane 1) or purified in peptidiscs (Lane 2).

DOI: https://doi.org/10.7554/eLife.46615.003

**Table 1.** List of GO terms used to predict protein association with the *E. coli* cell envelope.

**Gene ontology term (Associated with cell envelope)**

| | | | | | |
|---|---|---|---|---|---|
| Anchored component of membrane | Anchored component of external side of membrane | Anchored component of periplasmic side of outer membrane | Extrinsic component of periplasmic side of plasma membrane | Gram-negative bacterium cell wall | Extrinsic component of plasma membrane |
| Integral component of membrane | Cell envelope | Cell outer membrane | Integral component of cell outer membrane | Integral component of plasma membrane | Integral component of membrane |
| membrane | Cell wall | External side of cell outer membrane | Intrinsic component of cell outer membrane | Intrinsic component membrane | Intrinsic component of plasma membrane |
| Plasma membrane | Extrinsic component of cell outer membrane | Extrinsic component of membrane | Intrinsic component of external side of plasma membrane | Intrinsic component of periplasmic side of plasma membrane | Intrinsic component of periplasmic side of cell outer membrane |
| Outer-membrane bounded periplasmic space | Periplasmic space | Plasma membrane | Intrinsic component of cytoplasmic side of plasma membrane | Outer membrane | |
| Periplasmic side of outer membrane | Peptidoglycan-based cell wall | | | | |

DOI: https://doi.org/10.7554/eLife.46615.004

## Prediction of binary interactions and the high confidence subset

Binary protein interactions in the peptidisc library were predicted using PrInCE (Predicting Interactomes via Co-Elution), a software designed for analyzing PCP-SILAC datasets (*Stacey et al., 2017*). PrInCE predicts which protein pairs are interacting or not according to the similarity or dissimilarity of their fractionation profiles (*Figure 4A* and *Figure 4B*, respectively). As described in detail in the method section, a naive Bayes classifier was trained (10-fold cross-validation) using multiple pairwise similarity measures based on either the entire co-fractionation profile (Pearson correlation, Euclidean distance) and whether proteins shared an elution peak. To refine prediction, we also incorporated a single measure of pairwise similarity from the M3D expression database (*Faith et al., 2008*), since this allowed the classifier to distinguish between true interacting protein pairs and pairs whose fractionation profiles were only spuriously similar. Training labels for the classifier were generated from the gold standard complexes, with 'interacting' label applied to protein pairs in the same gold standard complex and 'non-interacting' label applied to pairs present in the gold standard list but not in the same gold standard complex. Using this approach, we predicted a list of 4911 pairwise interactions (*Figure 4C*; *Supplementary file 3*). These interactions were between protein pairs with well-correlated co-fractionation profiles (average R = 0.78 vs R = 0.16, respectively; Pearson correlation). The predicted list also captures the majority of gold standard pairwise interactions (recall = 0.80, *Figure 4D*).

Using our predicted peptidisc interactome, we then generated a High Confidence subset of interactions based on two orthogonal high-throughput interactomes collected independently from this study ('validating interactomes') (*Babu et al., 2018*). These High Confidence interactions were detected by three independent experiments in two laboratories. From the predicted peptidisc interactome list (4911 interactions), we identified 824 interactions also present in the two validating interactomes ('High Confidence' subset, *Supplementary file 6*). The extent of overlap is significantly greater than the number of overlapping interactions expected by chance (p<0.001, permutation test, *Figure 5—figure supplement 1A*). Consistent with the fact that high-throughput techniques are often biased toward detecting certain protein complexes over others (*Stacey et al., 2018*), our predicted peptidisc interactome has greater overlap with the co-fractionation validating interactome than the AP/MS validating interactome, although both overlaps are significant (N = 2382 and 1623, respectively, p<0.001 and p<0.001, *Figure 5—figure supplement 1B and C*).

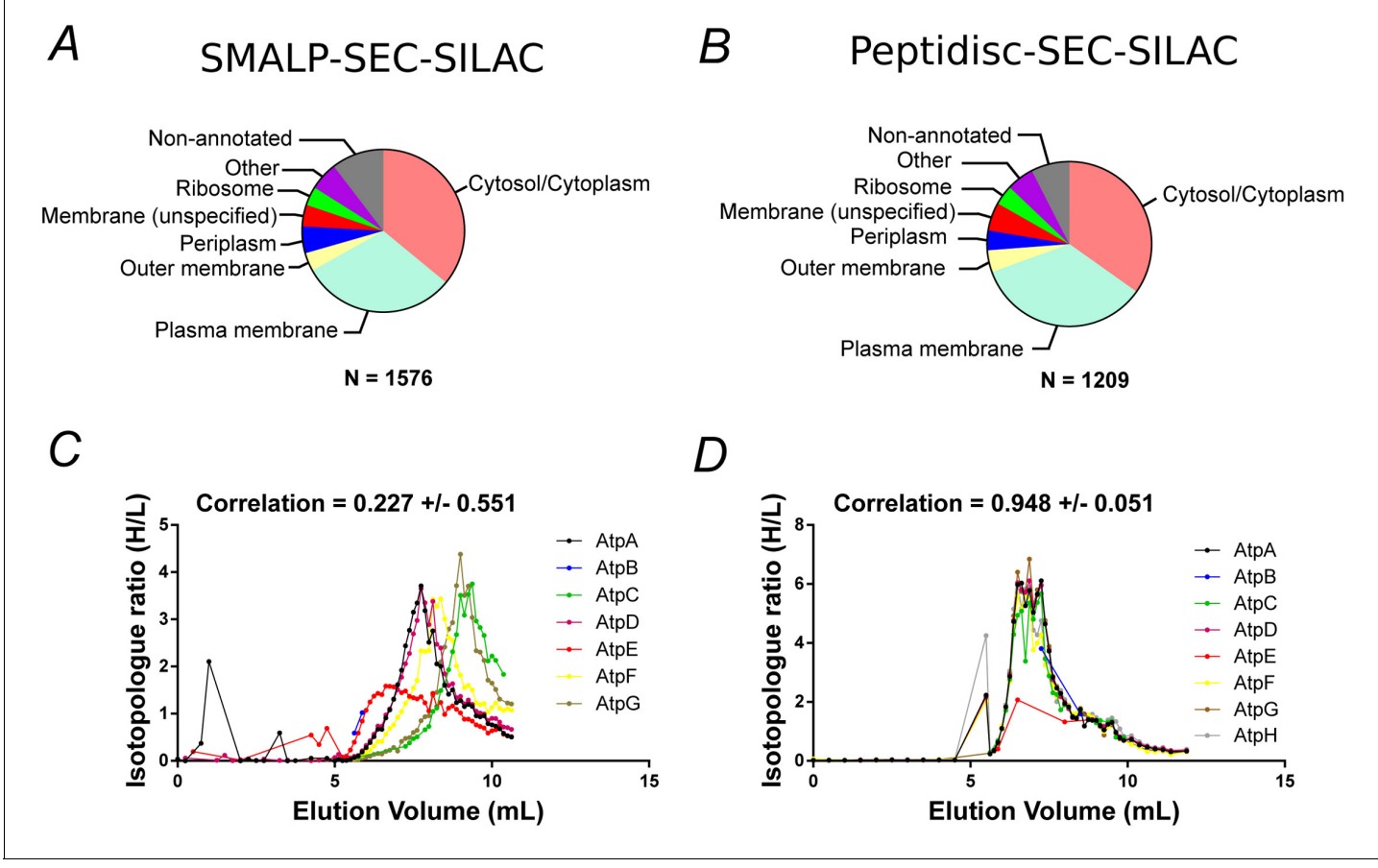

**Figure 3.** Proteomic analysis of soluble, SILAC-labeled *E. coli* membrane proteins in SMALPs or peptidisc libraries. Gene ontology analysis of identified proteins and annotated cellular compartment of identified proteins in (A) SMALP library, or in (B) peptidisc library. Co-fractionation profiles for quantified subunits of the ATP synthase complex in (C) SMALPs (raw data presented in *Supplementary file 2*) or (D) peptidisc (raw data presented in *Supplementary file 1*). Note: total volume of column is 18 mL and void volume is 6 mL. Void volume is represented as the zero on the x-axis of all co-elution graphs.

DOI: https://doi.org/10.7554/eLife.46615.005

The following figure supplement is available for figure 3:

**Figure supplement 1.** Fractionation profiles for select *E. coli* membrane protein complexes solubilized in SMA and peptidisc.

DOI: https://doi.org/10.7554/eLife.46615.006

## Computational validation of binary interactions

As false positives are inherent to high-throughput interactome studies, it is important to validate computationally that the interactome, on aggregate, indeed resembles a collection of true, biological interactors. To do so, we first used our gold standard protein complexes to calculate the ratio of TPs to FPs, measured as precision (TP / (TP + FP)) (*Figure 4C* and *Figure 4D*). However, the set of gold standard interactions in *E. coli* is relatively small compared to the set of gold standard interactions derived from mammalian studies (*Rajagopala et al., 2014*; *Ruepp et al., 2008*), meaning that any estimates based entirely on this gold standard set could be susceptible to noise. Therefore, to further estimate the biological plausibility of our peptidisc interactome, we determined whether predicted interacting pairs were more likely than non-interacting pairs to be enriched for three measures of biological association: shared GO terms, positively correlated stress phenotypes, and shared binding domains (*Erickson et al., 2017*; *Mosca et al., 2014*). Further, these enrichment values allowed us to benchmark our interactome against the *E. coli* cell envelope (CE) interactome recently published by *Babu et al. (2018)*.

Using these measures of biological plausibility, we found the peptidisc interactome to be enriched compared to random chance, and as shown in *Figure 5A–5C*, the enrichment is significant

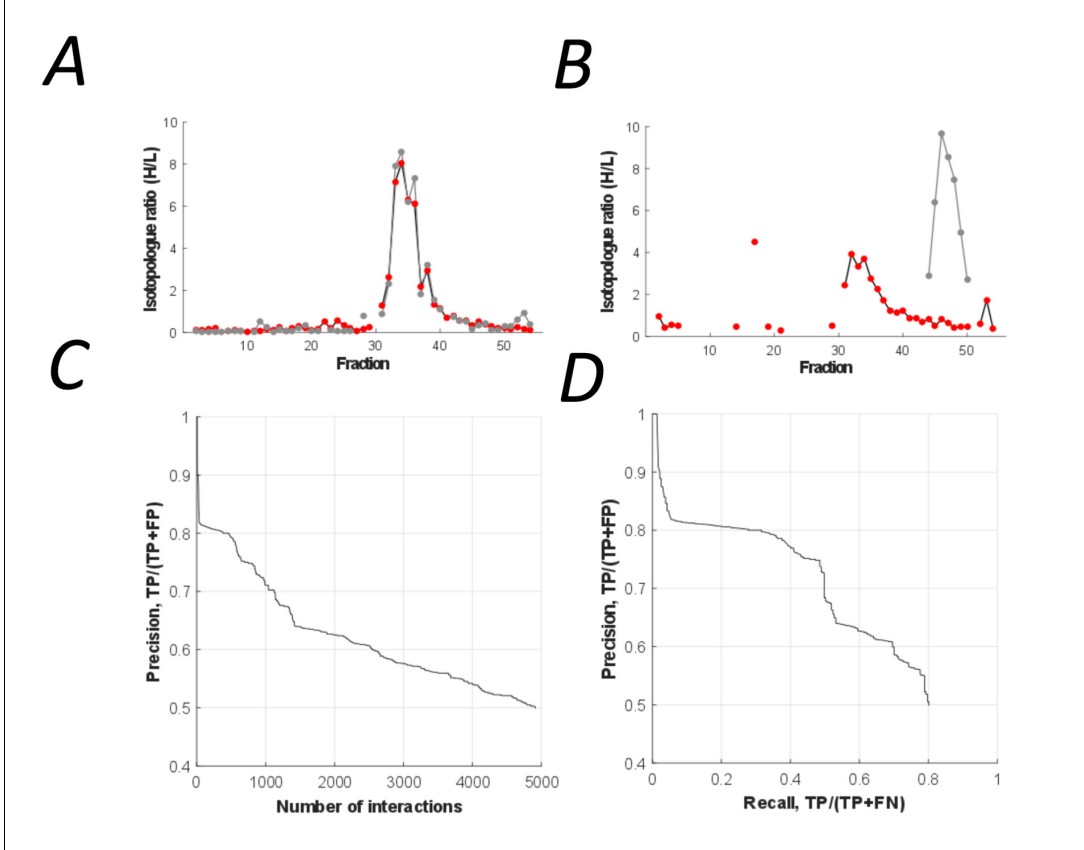

**Figure 4.** The peptidisc interactome is composed of 4911 co-fractionating protein pairs. (A) Typical elution profiles of an interacting protein pair in the peptidisc interactome. (B) Example elution profiles of non-interacting proteins. (C) Precision vs. accumulated number of interactions. (D) Precision-recall curve of the peptidisc interactome.

DOI: https://doi.org/10.7554/eLife.46615.007

for all three measures. Converting these measures to z-score, that is measuring relative to randomly rewired networks, our peptidisc interactome tended to be more enriched than the CE interactome (GO: z = 15.6 vs z = 14.2, peptidisc vs CE, respectively; Tolerome: z = 24.8 vs z = 6.2; 3did: z = 8.7 vs z = 6.3). We also calculated these three measures for our High Confidence set of interactions (*Figure 5*, black circles) and found that the High Confidence set was more enriched than the full peptidisc interactome for shared GO terms and positively correlated Tolerome profiles. The opposite was true for shared binding domains (*Figure 5C*), perhaps because of the sparsity of shared terms: only 24/824 interactions shared a binding domain in the High Confidence set (162/4911 full peptidisc interactome), indicating a noisier measure of biological association. In addition to benchmarking our enrichment values (GO, Tolerome, binding domains) against the CE interactome, we also confirmed that a significant number of interactions were common between the CE and peptidisc interactomes. Of the 4911 peptidisc interactions, 340 are also present in the CE interactome, a significant overlap (p<0.001, permutation test) (*Figure 5D*, *Supplementary file 6*). As expected, interactions that overlap with the CE interactome tend to be higher scoring than non-overlapping interactions (average interaction score 0.66 vs 0.62, respectively; p=5e-11, Wilcoxon rank-sum test; *Figure 4C*).

Finally, we confirmed that protein pairs in our peptidisc interactome had better-than-random M3D expression profile correlation (*Faith et al., 2008*) (*Figure 5E*). This is to be expected, since M3D expression correlation was used as a feature in our machine learning classifier (see Materials and methods), meaning high M3D correlation was a criterion on which our peptidisc interactions were selected. However, we also note that protein pairs in our peptidisc interactome had higher expression than protein pairs in the *E. coli* CE interactome (*Babu et al., 2018*). Therefore, as

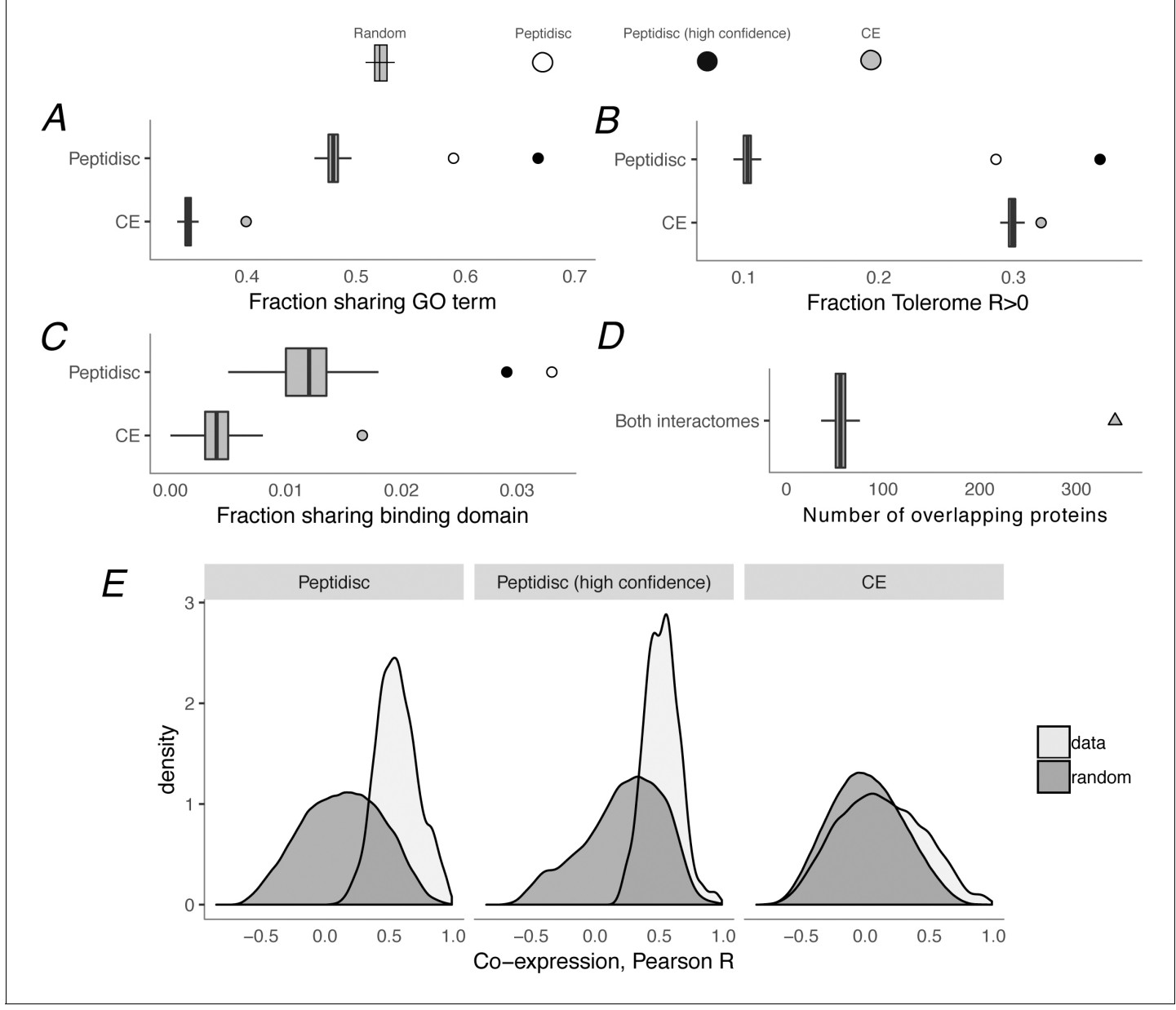

**Figure 5.** Computational validation of peptidisc interactome. (A) Fraction of interacting pairs sharing a gene ontology (GO) term for the peptidisc (top) and CE interactomes (bottom). Both the full peptidisc interactome (4911 interactions, white) and the High Confidence subset are shown (black). 'Random' shows the expected number of shared terms from randomly rewired peptidisc and CE interactomes (1000 iterations, gray bars). (B) Fraction of interacting pairs with positively correlated Tolerome profiles (R > 0, Pearson correlation). (C) Fraction of interacting pairs sharing binding domains. (D) Number of overlapping interactions between peptidisc and CE interactomes compared to random. (E) True ('data') and random distributions for M3D co-expression correlation (Pearson) for peptidisc, High Confidence, and CE. Random distributions generated by randomly rewiring networks.
DOI: https://doi.org/10.7554/eLife.46615.008

The following figure supplement is available for figure 5:

**Figure supplement 1.** Defining the High Confidence subset of interactions.
DOI: https://doi.org/10.7554/eLife.46615.009

expected, protein pairs in the peptidisc interactome were well-correlated as measured by M3D, and the level of correlation is higher than the benchmark CE interactome.

## Computational assignment and validation of protein complexes

We used a two-stage algorithm to cluster the identified pairwise interactions into complexes (*Wan et al., 2015*; *Drew et al., 2017*). A first stage clustering was performed using ClusterONE, an algorithm that allows moonlighting proteins present in multiple protein complexes (*Nepusz et al., 2012*). However, because ClusterONE tends to collapse biologically distinct protein groups into the same protein complex, we performed a second stage refinement using the MCL algorithm (*Enright et al., 2002*). The combination of these two algorithms ensured that the same protein can be assigned to multiple complexes. In addition, since both ClusterONE and MCL have tunable parameters, we performed a grid search optimization to find the parameter set which maximizes the matching ratio value between predicted complexes and our set of gold standard complexes. This procedure produced 202 complexes with a median size of five proteins (*Supplementary file 4*). As for the pairwise interactions above, we employed GO terms as an evidence for biologically meaningful complexes, and we reported that 36 of the 202 complexes were significantly enriched for at least one GO term (hypergeometric test, Benjamini-Hochberg-corrected $p<0.05$), a significant number ($p<0.001$, permutation test). Because clustering method removes pairwise interactions that are inconsistent with the predicted complexes, the subset of pairwise interactions clustered into complexes should be scoring higher than not clustered pairwise interactions (*Drew et al., 2017*). This was indeed the case: the 3490 pairwise interactions clustered in complexes had a significantly higher interaction score than the 1421 un-clustered interactions (mean interaction score 0.64 vs 0.59, $p=3e-72$, Wilcoxon rank-sum test).

## Experimental validation of binary interactions by affinity purification mass spectrometry (AP/MS)

Parallel to the in silico validation described above, we performed a series of in vitro experiments using three different membrane protein systems - the Sec translocon, the Bam complex and the ABC transporter MetNI. The goal was to use AP/MS to confirm and potentially discover novel pairwise associations predicted from interactome datalist. Principally, we aimed to confirm interactions between the core SecYEG complex and the membrane-anchored periplasmic chaperones YfgM and PpiD. These interactions are detected in our datalist at high (>75%) precision. However, this association is difficult to detect in detergent, unless all subunits are simultaneously over-produced in the membrane (*Figure 6—figure supplement 1*). We were also interested by the astonishing apparent interaction between the Sec and Bam complexes, also given in our datalist at high (>75%) precision. These include interactions between the SecY complex and the BamA, BamC and BamD subunits of the Bam complex. To perform these validation AP/MS experiments, the his-tagged SecYEG complex was expressed in SILAC labeling conditions. The membrane fraction was briefly solubilized with detergent followed by immediate trapping in peptidisc library. The SecY complex was subsequently isolated by Ni-NTA and the co-isolated proteins were identified by LC-MS/MS. To measure protein enrichment and to control for non-specific co-purifying background contaminants, the pulldown experiments were performed in parallel using a detergent extract or peptidisc library prepared from cells transformed with the empty vector.

Experiments with the detergent extract shows that SecY is highly enriched after affinity pulldown (*Figure 6A*, raw data presented in *Supplementary file 5*). There is also enrichment of several ribosomal proteins, which is not surprising given the intrinsic affinity of ribosomes for the Sec complex (*Rapoport et al., 2017*; *Park and Rapoport, 2012*). However, many known membrane-bound interactors of the SecY complex are not enriched, likely due to their dissociation during the prolonged incubation with detergent. Accordingly, in peptidisc, several ancillary subunits of the Sec translocon, including the periplasmic chaperones PpiD and YfgM, as well as the holo-translocon subunits YidC and YajC are detected. There is also strong enrichment of the porin OmpA in addition to several subunits of the outer membrane Bam complex. The BamB and BamC subunits in particular are significantly enriched (*Figure 6B*). The dynamics of a Sec-Bam complex interactions awaits further experimentation, but as it is this series of evidences validates the ability of the peptidisc PCP-SILAC and AP/MS workflows to capture novel protein assemblies that are difficult to isolate in detergent.

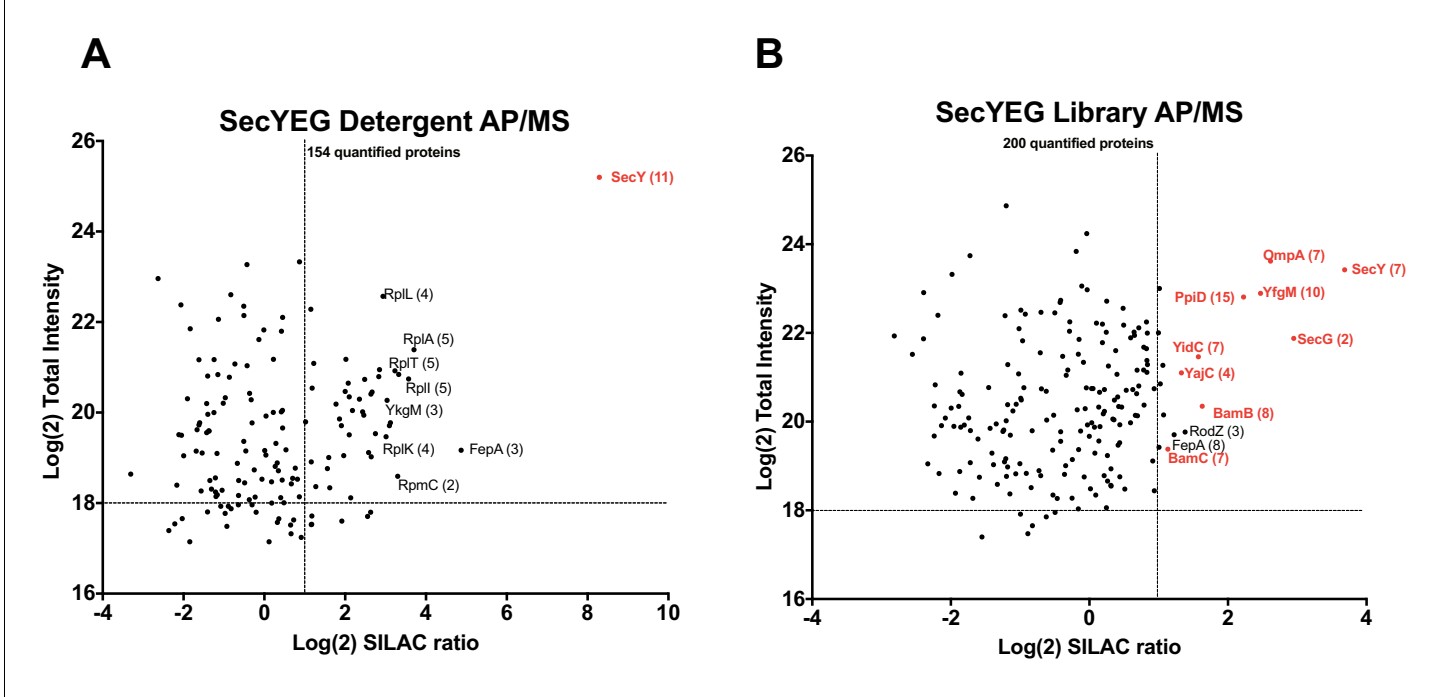

**Figure 6.** Validation of SecYEG interactors by AP/MS. (**A**) Enrichment matrix of each quantified protein identified in the SecYEG detergent AP/MS pulldown. The Log(2) peptide intensity for each quantified protein is plotted against the corresponding Log(2) SILAC ratio. Arbitrary enrichment cutoffs were set for both the x and y axes; these are indicated on the plot as dashed lines to aid the eye. Proteins of interest are highlighted in red. The number of unique peptides detected for each protein of interest is given in parentheses. Each black dot is a protein quantified in the pulldown experiments (**B**) As in A, but for proteins quantified in the SecYEG peptidisc library pulldown. Raw data for both plots are presented in *Supplementary file 5*.

DOI: https://doi.org/10.7554/eLife.46615.010

The following figure supplement is available for figure 6:

**Figure supplement 1.** Co-elution of SecYEG with YfgM_His-PpiD.

DOI: https://doi.org/10.7554/eLife.46615.011

## Identification of Bam complex interactors

We next applied the AP/MS workflow towards the protein BamA - the major subunit of the outer membrane-embedded Bam complex. Our interactome dataset identified the Sec ancillary subunits YidC and YajC as potential Bam interactors with high precision (>75%), as well as the cell surface-exposed lipoprotein RcsF (*Supplementary file 3*, *Figure 7A*). To explore the validity of these predicted interactions, we expressed his-tagged BamA in SILAC labeling conditions and analyzed the peptidisc library or detergent extract using the AP/MS workflow described above.

Experiments with the detergent extract shows that BamC and BamD are the only subunits enriched along with BamA (*Figure 7B*, raw data presented in *Supplementary file 5*). This finding is in agreement with an earlier study which showed that BamB is prone to dissociating from the rest of the complex in detergent solution (*Gu et al., 2016*). The only other interactor that is significantly enriched is RcsF. In peptidisc, by contrast, all four other subunits of the Bam complex (subunits B, C, D and E) are captured along with BamA (*Figure 7C*). Additionally, there is again significant enrichment of the Sec translocon ancillary subunits YidC and YajC, thereby providing additional evidence to support this potentially novel interaction. The lipoprotein RcsF and the porin OmpA are also significantly present (*Figure 7C*). We note that BamA-OmpA interaction was not reported in our interactome datalist, probably due to the unusually broad SEC elution profile of OmpA, leading to false negative identification due to low score precision. However, a series of recent publications have shown that OmpA is a *bona fide* interactor of both RcsF and BamA in the cell context (*Hart et al., 2019*; *Konovalova et al., 2014*).

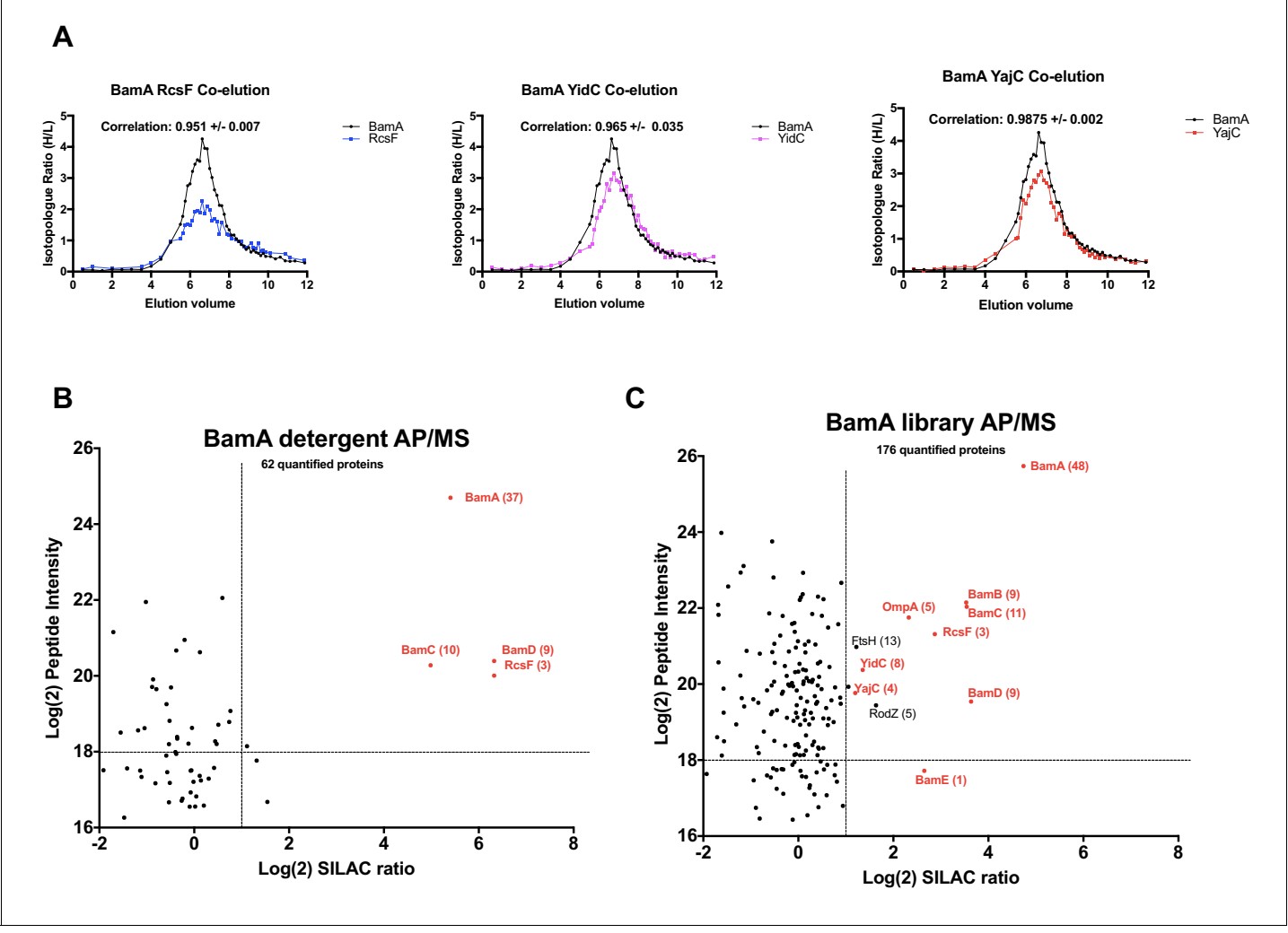

**Figure 7.** Validation of BamA interactors by AP/MS. (**A**) Pair-wise co-elution plots of select BamA interactors as predicted from the peptidisc PCP-SILAC workflow (raw data presented in *Supplementary file 1*). Pairwise interaction correlation values are shown above each plot. (**B**) Enrichment matrix of each quantified protein quantified in the BamA detergent AP/MS pulldown. The data was plotted and labeled as in *Figure 6*. (**C**) As in A, but for proteins quantified in the BamA peptidisc AP/MS pulldown. Raw data for the plots shown in B and C is presented in *Supplementary file 5*.
DOI: https://doi.org/10.7554/eLife.46615.012

## Identification of a unique type I ABC transporter complex

The ABC transporter MetNI was chosen as a third validation target because its PCP profile allowed to predict that *i*) MetQ forms a stable complex with MetNI and *ii*) NlpA (lipoprotein 28) is a novel interactor of the transporter (*Figure 8*). The stable interaction of MetQ with MetNI is unexpected because type I ABC transporters are characterized by weak affinity to their substrate binding proteins (SBPs). Accordingly, the transporters LivFGM and HisQP co-elute separately from the SBPs LivJ, LivK, and HisJ, respectively (*Figure 8—figure supplement 1*. Raw data presented in *Supplementary file 1*). In contrast, but as expected, the type II transporter FepC co-elutes tightly with FepB (*Bao and Duong, 2012*; *Bao and Duong, 2014*; *Hvorup et al., 2007*; *Rice et al., 2014*) (*Figure 8—figure supplement 1*). As an additional distinctive feature, we discovered the presence of lipobox at the N-terminus of MetQ, which is a unique case among other type I SBPs. This lipid modification could explain why MetQ remains associated to MetNI in the peptidisc library, and why this association has not been detected in earlier biochemical studies which employ the soluble, non-lipidated form of MetQ (*Nguyen, 2016*; *Nguyen et al., 2018*; *Nguyen et al., 2015*).

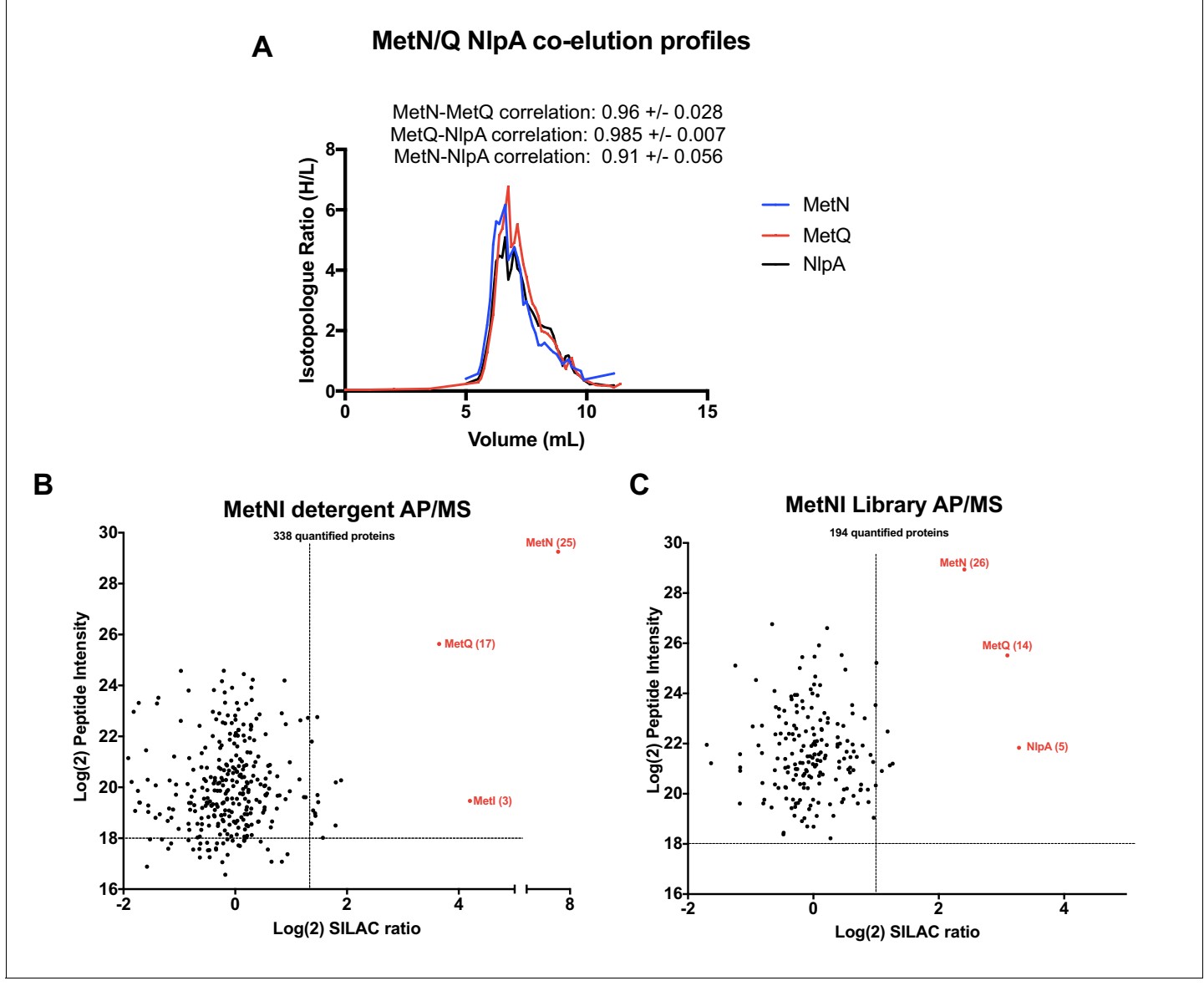

**Figure 8.** Validation of MetNI interactors of AP/MS. (A) Co-elution plot of MetN with the predicted interactors MetQ and NlpA (raw data presented in *Supplementary file 1*). Pairwise correlation values for each interaction are shown above the plot. (B) Enrichment matrix of each quantified protein quantified in the MetNI detergent AP/MS pulldown. The data was plotted and labeled as in *Figure 6*. (C) As in A, but for proteins quantified in the MetNI peptidisc AP/MS pulldown. Raw data for the plots shown in B and C is presented in *Supplementary file 5*.

DOI: https://doi.org/10.7554/eLife.46615.013

The following figure supplements are available for figure 8:

**Figure supplement 1.** Co-fractionation profiles of ABC transporters and SBPs in peptidisc.

DOI: https://doi.org/10.7554/eLife.46615.014

**Figure supplement 2.** Lipidation of MetQ is required for its interaction with the MetNI transporter.

DOI: https://doi.org/10.7554/eLife.46615.015

To validate these predictions, we expressed his-tagged MetNI in SILAC labeling conditions. As above, we isolated the MetNI complex in both detergent and peptidisc, and identified the co-purifying interactors using LC-MS/MS. The results reveal the interaction of MetQ with MetNI in both detergent and peptidisc (*Figure 8* - raw data presented in *Supplementary file 5*). The interaction of NlpA with MetNI is, however, detected only in peptidisc, and not in detergent (*Figure 8B and C*).

We next tested the importance of MetQ lipidation using a mutant carrying a disrupted lipobox, MetQ-C23A. The protein MetQ and MetQ-C23A were co-expressed with MetNI, the membrane fraction was solubilized in detergent and reconstituted in peptidisc. The his-tagged MetNI complex was subsequently isolated by Ni-NTA chromatography. The results indicate the co-elution of lipidated MetQ with MetNI but not MetQ-C23A, underscoring the importance of lipidation for this interaction (*Figure 8—figure supplement 2*).

## Discussion

Despite important progress in the field of proteomics, characterization of membrane interactomes has lagged due to the reliance on surfactants to maintain protein solubility. In this study, we show that the peptidisc can entrap membrane proteins directly out of a crude detergent membrane extract. The resulting protein library is stable, water soluble and is amenable to biochemical fractionation and characterization by mass spectrometry. There are distinctive advantages using peptidisc instead of detergents or other membrane mimetics in this experimental workflow. Because of its adaptability to membrane protein hydrophobic surfaces, the peptidisc is less likely compared to the nanodisc to bias reconstitution toward a certain protein diameter (*Bayburt et al., 2006*). Addition of exogenous lipids is also not required, which simplifies the reconstitution while reducing protein aggregation caused by extra lipids (*Roy et al., 2015*; *Wilcox et al., 2015*; *Marty et al., 2013*). The self-assembly is rapid and does not require dialysis or detergent adsorbents. This relative short exposure to surfactants is important as it minimizes detergent-mediated complex dissociation and aggregation. We further discuss the advantages and limitations of the peptidisc-proteomic workflow below, along with our initial biological findings using the *E. coli* membrane proteome as a model.

To compare the peptidisc against another membrane mimetic such as the SMA polymer, we measured the stability of three large membrane protein complexes, the ATP synthase, the Bam assembly and the respiratory chain I complex. Initial results revealed a high degree of overlap between the protein content of the libraries stabilized in peptidisc or in SMALPs (85.5% similarity), indicating that both methods effectively solubilize the membrane proteome (*Figure 3*). However, while the three large complexes listed above were largely intact in the peptidisc library, all three were significantly dissociated in the SMALPs library (*Figure 3*, *Figure 3—figure supplement 1*). Thus, the peptidisc method is demonstrably superior for capturing and stabilizing multi-subunit membrane protein complexes.

To determine the precision of the protein correlation profile (PCP) obtained with the peptidisc library, we had to separate a relatively small number of well-correlated protein pairs from a much larger background of non-interacting proteins. We used PrInCE, a machine learning bioinformatics tool written specifically for analyzing co-fractionation datasets (*Stacey et al., 2017*). This generated 4911 predicted interactions at 50% precision out of >700,000 potential random interactions. Like in every proteomic-based discovery method, we note the importance to benchmark the predicted interaction dataset against other databases (validating interactomes). As expected, our dataset was significantly enriched in multiple indicators of association, including gene ontology annotations, shared binding domains, and correlation of shared growth phenotypes. These indicators compare favorably with validating interactomes identified by low-throughput AP/MS, which generally have relatively few false positives (*Figure 5*; *Babu et al., 2018*). Importantly, a significant subset of interactions in our peptidisc interactome were also found within two other *E. coli* interactomes collected independently in a separate study (*Babu et al., 2018*). The integration of our peptidisc interactome with these validating interactomes lead to the 'High Confidence' dataset (*Supplementary file 6*). The number of interactions in this subset is significantly greater than that expected by chance (p<0.001, *Figure 5—figure supplement 1*), which supports the validity of the peptidisc workflow.

Parallel to this computational validation, we performed experiments to verify the ability of the peptidisc workflow to reveal novel or transient interactions. Our first validation target was the SecYEG complex. The pairwise peptidisc interaction list indicate a number of interactors at a high degree of precision (>75%; *Supplementary file 3*), such as the membrane-bound chaperones YfgM and PpiD. Interaction of SecY with these chaperones has been reported in the past using low-throughput 2D gel electrophoresis and cumbersome radio-labeling (*Götzke et al., 2014*). In our study, we show that combining the SecYEG complex trapped in peptidisc with the SILAC AP/MS workflow lead the facile detection of this interaction (*Figure 6*). In contrast, this association is hardly

seen in detergent unless all subunits are simultaneously over-produced in the membrane (*Figure 6—figure supplement 1*), which suggests that a proper subunit stoichiometry is critical for complex formation. This later observation highlights the importance of characterizing membrane protein networks of association under native expression conditions with minimal genetic manipulation.

Strikingly, the peptidisc interaction datalist also revealed a network of pairwise interactions between subunits of the Sec and Bam complexes. Interactions between SecYEG and the BamA, BamC and BamD subunits in particular are present in our interaction list at high precision. This Sec-Bam interaction has been suggested previously based on coincident detection between membrane fractionation and Western blotting (*Wang et al., 2016*). In order to confirm these interactions, the SecYEG complex was trapped in peptidisc and analyzed following the SILAC AP/MS workflow (*Figure 6B*). There is significant enrichment of the BamB and BamC subunits and these interactions are not detected when the same workflow is performed in detergent (*Figure 6A*). To obtain additional evidence for the Sec-Bam association, we employed BamA trapped in peptidisc as a bait using the same SILAC AP/MS workflow (*Figure 7*). Here also the results reveal compelling enrichment of the Sec ancillary subunits YidC and YajC, adding another layer of evidence for a Sec-Bam interaction. This interaction is particularly intriguing - and exciting - as it suggests a trans-membrane pathway for the direct transfer of proteins from the inner to outer membrane of the bacterial cell envelope. Further biochemical work is required to characterize this super-complex, but exciting new data already indicates the possibility to isolate the Sec/Bam assembly for structural analysis (http://dx.doi.org/10.1101/589077).

Other BamA interactors identified in the peptidisc-PCP-SILAC workflow were RcsF and OmpA (*Figure 7B*). The fact that the peptidisc is able to preserve these native interactions highlights the ability of the method to capture interactions that naturally exist in the cell membrane. Of note, the complex between RcsF, OmpA and BamA - which represents a novel mechanism of lipoprotein translocation to the extracellular side of the outer membrane - was initially revealed from genetic and in vivo cross-linking experiments, but never formally demonstrated by biochemical means (*Hart et al., 2019*; *Konovalova et al., 2014*). Accordingly, we find these interactions are much less apparent when the experiment was performed in detergent (*Figure 7A*).

The third validation target in this study is the transporter MetNI. We demonstrate that MetQ forms an unexpectedly stable complex with MetNI due of its N-terminal lipid anchor (*Figure 8*, *Figure 8—figure supplement 2*). The importance of MetQ lipidation in mediating this interaction has been overlooked in the literature to date and appears unique among amongst the 48 SBPs present in *E. coli*. This novel finding therefore calls for a re-evaluation of the role of the MetQ lipid anchor for complex stability and substrate transport. As an additional original discovery based on the peptidisc interactome data list, we identify and demonstrate the interaction of the MetNI complex with the lipoprotein NlpA (*Figure 8*). Earlier literature has reported that NlpA overproduction can complement a strain depleted for MetQ (*Zhang et al., 2003*), thereby providing a strong biological rationale to this interaction.

In conclusion, the peptidisc library combined with PCP-SILAC or AP/MS workflow is a promising novel approach for generating and validating high-throughput membrane protein interactomes. As the interaction list published here is intended to be a tool for future research, we provide two ways for researchers to narrow down the interactions to smaller subsets of interactions with fewer false positives. First, each interaction is given with an interaction score (*Supplementary file 3*), a measure which has been shown to correlate with measures of biological plausibility (*Stacey et al., 2017*). Second, the High Confidence datalist of interactions provides researchers with interactions that have been orthogonally validated (*Supplementary file 6*) (*Babu et al., 2018*). Further improvement of the precision to reduce the number of false positives is feasible by increasing the library fractionation, including for example separation over density gradient or ion exchange resins (*McBride et al., 2017*; *Maddalo et al., 2011*). However, this practice can also bias predicted interactions toward protein pairs that are already supported in the literature at the expense of detecting novel interactions. Looking forward, the peptidisc workflow can also be expanded to comparative analysis of membrane interactomes using a third amino acid isotopologue label. This labeling would allow profiling the changes in the global membrane protein interaction landscape under different conditions or in response to drugs and mutations.

## Materials and methods

### Reagents and plasmids

Tryptone, yeast extract, $Na_2HPO_4$, $KH_2PO_4$, NaCl, imidazole, Tris-base, acrylamide 40%, bis-acrylamide 2% and TEMED were obtained from Bioshop Canada. Amino acid isotopologues were purchased from Cambridge Isotope Laboratories. Isopropyl β-D-1-thiogalactopyranoside (IPTG), ampicillin, kanamycin, and arabinose were purchased from GoldBio. Detergents n-dodecyl-β-d-maltoside (DDM) and octyl-β-D-glucoside (β-OG) were from Anatrace. Detergent N,N-dimethyldodecylamine N-oxide (LDAO) was from Sigma. Columns Biosep 4000 GFC/SEC were purchased from Phenomenex. $Ni^{2+}$-NTA chelating Sepharose was obtained from Qiagen. Peptidisc peptides and biotinylated derivative Bio-Peptidisc (purity >90%) were obtained from Peptidisc Biotech Canada. All other chemicals were obtained from Fisher Scientific Canada. The genes *yfgM* or *ppiD* were inserted into pBAD33 encoding for a C-terminal 6x his-tag via the polymerase incomplete primer extension (PIPE) method (*Klock and Lesley, 2009*) to form pBad33-*yfgM* and pBad33-*ppiD*, respectively. To create pBad33-*YfgM-PpiD*, the sequence encoding for PpiD without a 6x his-tag was amplified and inserted into pBad33-*YfgM* using the PIPE method. The gene *BamA* was inserted into pBad22 by the PIPE method and 6x his-tag was subsequently inserted at the N-terminus of the mature protein. The plasmids pET19-_his_MetNI and pET21-MetQ_his_ were gifts from Dr. Janet Yang (University of San Francisco). Those plasmids were employed to construct pBad33-MetQ and pBad22-_his_MetNI. The *metQ* gene was amplified from pET21-MetQ_his_ and inserted into pBAD33. Plasmid pBAD22-_his_MetNI was constructed by sequentially inserting the *metN* and *metI* genes from pET19-_his_MetNI into pBAD22. The MetQ C23A mutation was inserted into pBad33-MetQ by site-directed mutagenesis. All construct sequences were confirmed by Sanger sequencing (Genewiz). The gene *msbA* was inserted with a sequence encoding for a N-terminal 6x his-tag into the vector pET28 to form the plasmid pET28-*msbA*. The plasmids pBad22-HA-EYG and pBad22-his-EYG have been previously described (*Tam et al., 2005*; *Maillard et al., 2007*; *Young and Duong, 2019*).

### Preparation of SILAC labeled *E. coli*

For preparation of heavy and light labeled crude membranes, *E. coli* strain JW2806 (*ΔlysA763::kan*) was labeled with Lys4 ($^2H_4$-lysine), as previously described (*Zhang et al., 2012*). Cells were picked from a single colony and grown overnight in 5 mL of LB + 25 µg/mL kanamycin at 37˚C. The overnight culture was isolated by low-speed centrifugation (5000 x g, 6 min), and resuspended in an equivalent volume of M9 minimal media. The culture was pelleted and washed two more times to ensure full removal of residual LB media. Unless otherwise stated, the cells were subsequently diluted 1/2000 into two flasks containing 250 mL M9 minimal media + 0.1% glucose + 100 µg/mL thiamine. The flasks were supplemented with either 0.06 mg/mL lysine or 0.06 mg/mL Lys4 to form the light- and heavy-labeled cultures, respectively. A control culture without supplemented amino acid was also inoculated but no growth was detected due to the inability of JW2806 to produce the essential amino acid lysine. Cells were grown at 37˚C for 16 hr until OD reached ≅ 0.9–1.1.

### Optimization of the peptidisc library method using the model membrane protein MsbA

Plasmid pET28-_his_MsbA was expressed in *E. coli* BL21(DE3) (New England Biolabs) for 3 hr at 37˚C after induction with 0.5 mM IPTG at an OD of 0.4–0.7 in LB medium supplemented with 25 µg/mL kanamycin. Cells were harvested by low speed centrifugation (10,000 x g, 6 min) and resuspended in SEC buffer (50 mM Tris-HCl, pH 7.2; 50 mM Na-acetate; 50 mM K-acetate). Resuspended cells were treated with 1 mM phenylmethylsulfonyl fluoride (PMSF) and lysed using a french press at 10,000 psi. Unbroken cell debris and other aggregates were removed by an additional low-speed centrifugation. The crude membrane fraction containing overexpressed MsbA was subsequently isolated by ultracentrifugation (100,000 x g, 45 min). The crude membrane fraction was resuspended in SEC Buffer to a concentration of ~20 mg/mL. To screen different detergents, aliquots of MsbA-containing crude membranes were solubilized in either 1% DDM, 3% β-OG, 1% DOC or 1% LDAO. Solubilizations were performed at 4˚C for 1 hr. Insoluble material was then pelleted by ultracentrifugation (100,000 x g, 15 min). The detergent-solubilized extracts were subsequently trapped in peptidisc libraries as described below. To compare the efficiency of library capture between the different

detergents assayed, aliquots of each detergent extract and each resultant peptidisc library were analyzed side by side on 15% SDS-PAGE followed by Coomassie Blue staining. Peptidisc-MsbA was subsequently isolated by Ni$^{2+}$-chelating chromatography in SEC buffer, washed in 10 column volumes (CV) of Wash Buffer (20 mM Tris-HCl: pH 7.1; 50 mM K-acetate; 50 mM Na-acetate; 15 mM imidazole), and then eluted in ½ CV Elution Buffer (20 mM Tris-HCl: pH 7.1; 50 mM K-acetate; 50 mM Na-acetate; 400 mM imidazole). For purification of MsbA in DDM, the procedure was repeated except there was no addition of Peptidisc peptide to the solubilized crude membrane and a concentration of 0.02% DDM was maintained in all buffers during the dilution and purification steps.

## Incorporation of *E. coli* cell envelope proteins in peptidisc library

Cells grown to OD ~0.9–1.2 were pelleted by low-speed centrifugation (5000 x g, 6 min) and resuspended in 2 mL Buffer A. Cells were lysed by French press (10,000 psi, two passages) and cell debris removed by an additional low-speed centrifugation step (10,000 x g for 10 min). Crude membrane was isolated by ultracentrifugation (100,000 x g, 45 min), and resuspended in Buffer A to a protein concentration of 20 mg/mL. The crude membrane was solubilized in 0.8% DDM, and isolated by ultracentrifugation. Solubilized crude membrane (100 µL at 10 mg/mL) was mixed with the Bio-Peptidisc peptide (350 µL at 6 mg/mL), and the mixture diluted to 10 mL ([DDM] $\cong$ 0.008%). The mixture was concentrated over a 100 kDa cut-off polysulfone filter (Sarstedt) to 500 µL, then diluted again to 5 mL in Buffer A ([DDM] $\cong$ 0.0008%). The library was concentrated to 250 µL ([Total protein] $\cong$ 6 mg/mL) and left on ice until fractionation. For pull-down experiments, the libraries were concentrated to $\cong$1 mg/mL and placed on ice until subsequent use.

## Incorporation of *E. coli* cell envelope proteins in SMALP

The SMA polymer containing 2:1 styrene to maleic acid ratio was prepared following the procedure described in reference (*Dörr et al., 2014*). In brief, 10% of SMA 2000 (Cray Valley), was refluxed for 3 hr at 80°C in 1M KOH, resulting in complete solubilization of the polymer. Polymer was then precipitated by dropwise addition of 6M HCl accompanied by stirring and pelleted by centrifugation (1500 x g for 5 min). The pellet was then washed 3 times with 50 mL of 25 mM HCl, followed by a third wash in ultrapure water and subsequent lyophilization. SMA (pre- and post-hydrolysis) was analyzed by Fourier Transform-Infrared Spectroscopy (FT-IR) to confirm full hydrolysis of the anhydride group. The hydrolyzed SMA was later re-suspended at 10% wt/vol in 25 mM Tris-HCl, and the pH of the solution adjusted to 8.0 with 1M NaOH. Cells were pelleted by low-speed centrifugation (5000 x g, 6 min), and resuspended in 2 mL SEC Buffer. Cells were lysed by french press (10,000 psi, two pass) and cell debris removed by an additional low-speed centrifugation step (10,000 x g, 10 min). Crude membrane was isolated by ultracentrifugation (100,000 x g, 45 min), and resuspended in SEC Buffer to a protein concentration of 20 mg/mL. Crude membranes were solubilized by addition of 3% SMA2000 for 1 hr at 4°C, clarified by ultracentrifugation (100,000 x g, 15 min, 4°C), then placed on ice until subsequent use.

## Fractionation of cell envelope libraries and digestion of protein samples

Cell envelope protein libraries were fractionated by size exclusion chromatography as previously described (*Kristensen et al., 2012*; *Zhang et al., 2012*; *Scott et al., 2017*). In brief, 200 µL of prepared libraries were separated over two tandem BioSep4000 columns (Phenomenex) pre-equilibrated in SEC buffer at 8°C. At an isocratic flow of 0.5 mL/min, fractions were collected from 20 min to 44 min. After fractionation - where applicable - detergent was first removed from protein samples by acetone precipitation. In brief, protein sample was mixed with 80% ice cold acetone, then left overnight on ice to precipitate. The precipitated proteins were pelleted by low-speed centrifugation (10,000 x g, 10 min, 4°C), washed with an equivalent volume of ice cold, 100% acetone and pelleted again (10,000 x g, 10 min, 4°C). The supernatant was aspirated away and pellet air-dried at 42°C for 10 min before storage at −20°C until digestion. For peptidisc libraries, detergent was removed during peptidisc assembly, so no acetone precipitation was necessary. We used a modified protocol to digest protein samples into tryptic peptides (*Scott et al., 2017*). In brief, samples were first denatured in 6M urea. When Bio-Peptidisc peptide was present in the sample, the denatured proteins were incubated with streptavidin coated agarose beads (2 µg beads/µl pre-washed in SEC Buffer) for 30 min at 25°C and the supernatant removed to deplete the peptide. Denatured proteins were

incubated with 5 mM DTT for 1 hr at 25℃ to reduce any cysteines. Free cysteines were alkylated by addition of 20 mM iodoacetamide for 1 hr at 25℃ in the dark, the reaction was then quenched by addition of 40 mM DTT. Samples were pre-cleaved by addition of 0.1 µg Lys-C for 1.5 hr at 25℃, followed by dilution to 1 M urea in 50 mM ammonium bicarbonate, pH 8.3. Proteomics grade trypsin (1 µg; Promega) was added to each sample, and the reactions left to digest overnight at 25℃. Digested samples were acidified to <pH 2.5 by addition of 1% trifluoroacetic acid and the resulting peptide supernatant purified using self-made Stop-and-go-extraction tips (StageTips) composed of C18 Empore material (3M) packed in to 200 µl pipette tips (*Ishihama et al., 2006*; *Rappsilber et al., 2003*; *Rappsilber et al., 2007*). Prior to addition of the peptide solution, StageTips were conditioned with methanol and equilibrated with 0.5% acetic acid (Buffer A3). Peptide supernatants were loaded onto columns and washed with three bed volumes of Buffer A3. Peptide samples were eluted with 80% acetonitrile, 0.5% acetic acid (Buffer B3) into microfuge tubes, dried down using a vacuum concentrator, and stored at −20℃.

### Expression of the validation targets SecYEG, MetNI and BamA in SILAC labeling conditions

Plasmids pBad22, pBad22-hisEYG, pBad22-hisMetNI and pBad22-hisBamA were chemically transformed into *E. coli* JW2806. Cells were grown overnight in M9 media supplemented with either 0.3 mg/mL Lys4 (for pBad22-hisEYG, pBad22-hisMetNI and pBad22-hisBamA) or 0.3 mg/mL light Lysine (for pBAD22). The next morning, the cultures were diluted 1/100 into fresh M9 media supplemented with either Lys4 or light lysine. Protein expression was induced with 0.02% arabinose once the cells had reached OD ~0.4–0.6. The cultures were then shifted to 25℃ and grown overnight. Cells were harvested and resuspended in TSG buffer containing 1 mM PMSF before being lysed on a French Press (8000 psi, three passes). The membrane fraction was collected and resuspended in TSG (50 mM Tris HCl pH 8; 50 mM NaCl; 10% glycerol) buffer, rather than in SEC buffer. Membranes were solubilized in 0.5% DDM for 15 min on ice. Solubilized material was clarified by ultracentrifugation (100,000 x g, 15 min). An aliquot of the detergent-solubilized material was purified as described above, except that all steps contained TSG buffer, with DDM and imidazole where necessary. The remainder of the detergent-soluble supernatant was mixed with a 4:1 excess of Peptidisc peptide and peptidisc libraries were prepared by the dilution and concentration method described above. The resultant library (~1 mL at 1 mg/mL) was purified as described above for MsbA, except that all steps contained TSG buffer (with imidazole when necessary), not SEC buffer. Eluted proteins were analyzed by 15% SDS-PAGE and visualized by Coomassie Blue staining. For mass spectrometry analysis, the detergent-purified 'heavy' and 'light' elutions fractions were pooled and acetone precipitated before being digested with trypsin and LysC. The samples were then STAGE tipped and analyzed by mass spectrometry. The 'heavy' and 'light' peptidisc elution fractions were pooled, denatured with 6M urea and digested with trypsin and LysC before STAGE tipping and analysis by mass spectrometry.

### Validation of the YfgM-PpiD-SecYEG interaction

Plasmids pBad33-*YfgM*, pBad33-*PpiD*, and pBad33-*YfgM PpiD* were transformed into chemically competent BL21DE3 cells harboring the plasmid pBad22-*HA-EYG.* For expression of his-tagged SecYEG complex only, plasmid pBad22-*his-EYG* was transformed into BL21DE3 cells. Overnight cultures were prepared in LB media supplemented with appropriate antibiotics at the concentrations specified above. After an overnight incubation, the cultures were diluted 1:100 into fresh LB media with antibiotics. Protein expression was induced at OD ~0.4–0.6 by addition of 0.1% arabinose, and cultures were grown for a further 2 hr before harvesting. Cells were resuspended in TSG buffer before being lysed as described above. Membranes were prepared in TSG buffer and solubilized as described above. Solubilized material was clarified by ultracentrifugation (100,000 x g, 15 min) before incubation for 30 min with Ni-NTA affinity resin. Beads were washed in 10 CV TSG buffer + 0.02% DDM, then eluted in TSG buffer + 300 mM imidazole + 0.02% DDM. Eluted proteins were analyzed by 15% SDS-PAGE followed by either Coomassie Blue staining or a western blot using a SecY-specific antibody as previously described (*Dalal and Duong, 2009*; *Dalal et al., 2012*).

## Validation of the MetNI-Q interaction

Plasmids pBad33-MetQ and pBad33-MetQ C23A were transformed into chemically competent BL21DE3 cells containing the plasmid pET21-$_{his}$MetNI. Overnight cultures were prepared in LB media supplemented with appropriate antibiotics. After an overnight incubation, the cultures were diluted 1:100 into fresh LB media with antibiotics. Protein expression was induced at OD ~0.4–0.6 by addition of 0.1% arabinose and 1 mM IPTG, and cultures were grown for a further 2 hr before harvesting. Cells were resuspended in TSG buffer (50 mM Tris-HCl pH 8; 50 mM NaCl; 10% glycerol), and the membrane fraction was prepared and solubilized as described above. The solubilized material (~1 mL) was mixed with a 4:1 excess of Peptidisc peptide, and peptidisc libraries were prepared by the dilution and concentration method described above. The resultant library (~1 mL at 1 mg/mL) was incubated with Ni-NTA resin for 30 min at 4°C. The resin was washed with 10 CV of TSG buffer, then eluted in TSG buffer + 300 mM Imidazole. Eluted proteins were analyzed by 15% SDS PAGE and visualized by Coomassie Blue staining.

## Liquid chromatography and mass spectrometry analysis

Purified peptides were analyzed using a quadrupole – time of flight mass spectrometer (Impact II; Bruker Daltonics) on-line coupled to an Easy nano LC 1000 HPLC (ThermoFisher Scientific) using a Captive spray nanospray ionization source (Bruker Daltonics) including a 2-cm-long, 100 μm-inner diameter fused silica fritted trap column, 75 μm-inner diameter fused silica analytical column with an integrated spray tip (6–8 μm diameter opening, pulled on a P-2000 laser puller from Sutter Instruments). The trap column is packed with 5 μm Aqua C-18 beads (Phenomenex, www.phenomenex. com) while the analytical column is packed with 1.9 μm-diameter Reprosil-Pur C-18-AQ beads (Dr. Maisch, www.Dr-Maisch.com). Buffer A consisted of 0.1% aqueous formic acid in water, and buffer B consisted of 0.1% formic acid in acetonitrile. Samples were resuspended in buffer A and loaded with the same buffer. Standard 45 min gradients were run from 0% B to 35% B over 90 min, then to 100% B over 2 min, held at 100% B for 15 min. Before each run the trap column was conditioned with 20 μL buffer A, the analytical – with 4 μL of the same buffer and the sample loading was set at 20 μL. When one column system was used the sample loading volume was set at 8 μL + sample volume. The LC thermostat temperature was set at 7°C. The Captive Spray Tip holder was modified similarly to an already described procedure (*Beck et al., 2015*) – the fused silica spray capillary was removed (together with the tubing which holds it) to reduce the dead volume, and the analytical column tip was fitted in the Bruker spray tip holder using a piece of 1/16' x 0.015 PEEK tubing (IDEX), an 1/16' metal two way connector and a 16–004 Vespel ferrule. The sample was loaded on the trap column at 850 Bar and the analysis was performed at 0.25 μL/min flow rate. The Impact II was set to acquire in a data-dependent auto-MS/MS mode with inactive focus fragmenting the 20 most abundant ions (one at the time at 18 Hz) after each full-range scan from m/z 200Th to m/z 2000Th (at 5 Hz rate). The isolation window for MS/MS was 2 to 3Th depending on parent ion mass to charge ratio and the collision energy ranged from 23 to 65 eV depending on ion mass and charge (*Beck et al., 2015*). Parent ions were then excluded from MS/MS for the next 0.4 min and reconsidered if their intensity increased more than five times. Singly charged ions were excluded since in ESI mode peptides usually carry multiple charges. Strict active exclusion was applied. Mass accuracy: error of mass measurement is typically within five ppm and is not allowed to exceed 10 ppm. The nano ESI source was operated at 1900V capillary voltage, 0.20 Bar CaptiveSpray nanoBooster pressure, 3 L/min drying gas and 150°C drying temperature.

Analysis of Mass Spectrometry Data was performed using MaxQuant 1.5.3.30 (*Cox and Mann, 2008*; *Cox et al., 2014*; *Tyanova et al., 2014*). The search was performed against a database comprised of the protein sequences from the source organism (*E. coli* K12) plus common contaminants using the following parameters: peptide mass accuracy 40 parts per million; fragment mass accuracy 0.05 Da; trypsin enzyme specificity, fixed modifications - carbamidomethyl, variable modifications - methionine oxidation, deamidated N, Q and N-acetyl peptides. Proteins were quantified from one peptide identification. Only those peptides exceeding the individually calculated 99% confidence limit (as opposed to the average limit for the whole experiment) were considered as accurately identified.

## Binary interaction prediction using PrinCE

Protein-protein interactions were predicted using PrInCE (*Stacey et al., 2017*), a co-fractionation data analysis pipeline that assigns interactions based on the similarity of co-fractionation profiles. The PrInCE software is available online: https://github.com/fosterlab/PrInCE-Matlab. Since PrInCE employs a naive Bayes classifier, a set of known interacting and non-interacting protein pairs are required to train the classifier, that is a gold standard set. We constructed a gold standard set of protein complexes by combining the 30S ribosome with membrane protein complexes given by the IntAct protein complex database (www.ebi.ac.uk/complexportal/). True positive interactions (TP) are between proteins present in the same gold standard complex, and false positive interactions (FP) are interactions between proteins present in the gold standard set but which are not members of the same complex. PrInCE calculates an interaction score for each protein pair, with higher scores indicating an interaction is more likely to be a true interaction, as measured by the proportion of gold standard TPs. Specifically, a protein pair's interaction score is equal to the TP-to-FP ratio, measured as precision (TP/(TP +FP)) of all predicted interactions with a classifier score greater than that protein pair. Both the precision of the full list (50%) and the interaction score are directly related to an interaction false discovery rate (FDR), as FDR = 100% precision. For full implementation of PrInCE see *Stacey et al. (2017)*.

Unlike other co-fractionation analyses, which associate protein pairs using external datasets such as gene co-citation (*Larance et al., 2016*), PrInCE is designed to use only information derived from the experimental dataset. Using external datasets for interaction prediction can bias results to well-known, highly studied interactions (*Skinnider et al., 2018*). However, since there are still a considerable number of annular lipids retained in peptidiscs, the molecular weight of protein complexes can vary and broaden elution peaks (*Carlson et al., 2018*) and thereby increase false positives. Therefore, we struck a balance between predicted interactome size and interaction novelty by including a single external dataset, the M3D database (Many Microbes Microarray Database; *Faith et al., 2008*). For each protein pair observed in our experimental dataset, we calculated the Pearson correlation between expression profiles from M3D. Protein pairs not in the M3D database were imputed as the mean correlation value.

## Protein complex assignment via ClusterONE and MCL

We used a two-stage procedure to cluster pairwise interactions into complexes (*Wan et al., 2015*; *Drew et al., 2017*). A first pass clustering was performed using ClusterONE (*Nepusz et al., 2012*), the results of which were further refined using the Markov Cluster algorithm (MCL) (*Enright et al., 2002*). This procedure resulted in clustered and unclustered protein interactions. In order to incorporate unclustered but high-scoring interactions (interaction score >0.75), we re-ran the two-stage clustering (ClusterONE + MCL) using unclustered pairwise interactions with score >0.75 as input. The union of these two sets of clusters formed the final set of complexes. Using both ClusterONE and MCL ensured that the same protein can be assigned to multiple complexes, while avoiding collapsing biologically distinct protein groups into the same protein complex, which ClusterONE tends to do (*Wan et al., 2015*). We measured clustering performance using the maximum matching ratio, a score calculated on the best one-to-one mapping between predicted and gold standard complexes (*Nepusz et al., 2012*).

Since both ClusterONE and MCL have tunable parameters, we performed a grid search optimization to find the parameter set which maximized the matching ratio value. The optimized parameters were: *p*, a ClusterONE parameter that models incompleteness in the network by assuming the existence of interactions outside of the given network; *dens*, a ClusterONE parameter that controls the minimum density of complexes; *I*, the single MCL parameter, which controls the granularity of MCL output; and *S*, which is the minimum interaction score of pairwise interactions fed into the clustering algorithm. The optimized values were p=*5000*, *dens = 0.001*, *I = 20*, and *S = 0.50* (equal to 50% precision, that is the entire peptidisc interactome).

## Computational validation of binary interactions and protein complexes

We performed multiple validations of both binary interactions and protein complexes using custom Matlab code. To validate the binary interactions, we first calculated three measures of biological association: *i*) fraction of protein pairs sharing at least one Gene Ontology (GO) term, *ii*) fraction of

protein pairs with a positively correlated stress phenotypes as measured in the Tolerome database (R > 0, Pearson correlation; *Erickson et al., 2017*), and *iii*) fraction of protein pairs sharing at least one three-dimensional interacting domains (3did) (*Mosca et al., 2014*). Only GO and 3did terms that annotated >20 and <1000 proteins were used. Null distributions for each measure were calculated by generating 1000 random peptidisc interactomes, each composed of 4911 random, unique interactions between proteins in the peptidisc interactome. Each of the three measures of biological association were calculated for the 1000 random random interactomes. Z-scores were calculated relative to these null distributions.

We also calculated whether the number of overlapping interactions between our binary interaction list and a recently published *E. coli* cell envelope (CE) interactome (*Babu et al., 2018*) was significant. This was calculated by randomly assigning 4911 unique interactions to the set of proteins participating in our interactions. By calculating the overlap between (*Babu et al., 2018*) and random interactomes, we estimated the probability that the true number of overlapping interactions was due to chance alone. To validate complexes, we calculated the number of complexes enriched for at least one GO term (hypergeometric test, Benjamini-Hochberg correction). To obtain significance for the number of enriched complexes, we repeated this enrichment analysis with 1000 sets of random complexes, where each set was composed of 227 complexes and the size distribution was preserved from the original predicted complexes. Each random complex was generated by randomly sampling from the 526 proteins participating in the predicted complexes. GO terms were filtered such that only terms assigned to >20 and <1000 proteins were used.

As an additional computational validation, we determined the subset of our interactome that is also detected by two independent high-throughput interactomes. These validating interactomes are *i*) a detergent-solubilized size exclusion co-fractionation interactome (78984 interactions) and *ii*) an AP/MS dataset (499605 interactions). For methods of preparation, see *Babu et al. (2018)*. These validating interactomes were collected independently from the dataset used to generate our peptidisc interactome and therefore provide orthogonal validation. Together, the set of pairwise interactions that were common to all three datasets (the peptidisc interactome and both validating interactomes) form our 'High Confidence' set of interactions.

## Additional information

### Competing interests

Franck Duong Van Hoa: has a website which sells the peptide used in this study. The other authors declare that no competing interests exist.

### Funding

| Funder | Author |
|---|---|
| Canadian Institutes of Health Research | Mohan Babu Leonard J Foster Franck Duong Van Hoa |

The funders had no role in study design, data collection and interpretation, or the decision to submit the work for publication.

### Author contributions

Michael Luke Carlson, Conceptualization, Data curation, Formal analysis, Investigation, Visualization, Methodology, Writing—original draft; R Greg Stacey, Conceptualization, Software, Formal analysis, Validation, Investigation, Visualization, Methodology, Writing—original draft, Writing—review and editing; John William Young, Conceptualization, Data curation, Formal analysis, Validation, Investigation, Visualization, Methodology, Writing—original draft, Writing—review and editing; Irvinder Singh Wason, Zhiyu Zhao, Data curation, Investigation, Visualization, Methodology; David G Rattray, Data curation, Investigation; Nichollas Scott, Conceptualization, Data curation, Investigation, Methodology; Craig H Kerr, Data curation, Formal analysis, Investigation, Methodology; Mohan Babu, Data curation, Formal analysis, Writing—review and editing; Leonard J Foster,

Conceptualization, Resources, Supervision, Funding acquisition; Franck Duong Van Hoa, Conceptualization, Resources, Formal analysis, Supervision, Funding acquisition, Writing—original draft, Project administration, Writing—review and editing

### Author ORCIDs
Michael Luke Carlson https://orcid.org/0000-0002-3807-6516
R Greg Stacey https://orcid.org/0000-0002-4496-8131
John William Young https://orcid.org/0000-0003-3541-509X
Nichollas Scott http://orcid.org/0000-0003-2556-8316
Mohan Babu http://orcid.org/0000-0003-4118-6406
Leonard J Foster http://orcid.org/0000-0001-8551-4817
Franck Duong Van Hoa https://orcid.org/0000-0001-7328-6124

### Decision letter and Author response
Decision letter https://doi.org/10.7554/eLife.46615.024
Author response https://doi.org/10.7554/eLife.46615.025

## Additional files

### Supplementary files
• Supplementary file 1. Protein enrichment in SEC fractionated peptidiscs (Biological Replicates 1 and 2).
DOI: https://doi.org/10.7554/eLife.46615.016

• Supplementary file 2. Protein enrichment in SEC fractionated SMALPs (Biological Replicates 1 and 2).
DOI: https://doi.org/10.7554/eLife.46615.017

• Supplementary file 3. Annotated, binary interaction list of membrane proteins identified in the peptidisc library by PCP-SILAC.
DOI: https://doi.org/10.7554/eLife.46615.018

• Supplementary file 4. Predicted complexes from identified binary interactions in peptidisc library at 50% precision.
DOI: https://doi.org/10.7554/eLife.46615.019

• Supplementary file 5. Enrichment of proteins in AP/MS pulldowns of SecYEG, BamA, and MetNI.
DOI: https://doi.org/10.7554/eLife.46615.020

• Supplementary file 6. 'High Confidence' interaction list - Protein interactions detected in both this study and *Babu et al., 2018*.
DOI: https://doi.org/10.7554/eLife.46615.021

• Transparent reporting form
DOI: https://doi.org/10.7554/eLife.46615.022

### Data availability
All data generated or analyzed during this study are included in the manuscript and supporting files.

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
