## [Decision Letter]

[Editors’ note: a previous version of this study was rejected after peer review, but the authors submitted for reconsideration. The first decision letter after peer review is shown below.]

Thank you for submitting your work entitled "Profiling the *E. coli* Membrane Interactome in Detergent-free Peptidiscs" for consideration by *eLife*. Your article has been reviewed by three peer reviewers, and the evaluation has been overseen by a Reviewing Editor and a Senior Editor. The following individual involved in review of your submission has agreed to reveal their identity: Carol V. Robinson (Reviewer #1).

Our decision has been reached after consultation between the reviewers. Based on these discussions and the individual reviews below, we regret to inform you that your work will not be considered further for publication in *eLife*. Amongst the issues raised by the reviewers, they particularly emphasised the need for better treatment of false positive identifications.

Reviewer #1:

The manuscript describes a new approach for extracting and maintaining interactions in peptidiscs. This a very interesting paper worthy of publication in *eLife*. The study is complementary, yet distinct from nanodiscs and SMALPS, and appears to have some unique advantages.

Although the membrane proteome is still extracted with detergent prior to reconstitution results are similar to those obtained in SMALPs and nanodiscs. The number of unique insights obtained through the combination of methods applied here was however difficult to tease out. It seems that the most important one was that MetQ co-elutes as a complex with MetNI. MetQ appears to be the only substrate binding protein that is lipid modified at its N-terminus. This I thought particularly interesting.

Of the SecYEG translocon interacting partners it as surprising to me that simultaneous over- production of the SecYEG core translocon with both PpiD and YfgM is necessary for detection of this interaction. It might suggest that original interactions which are thought to be multiple are disrupted by the initial detergent solubilisation step. Perhaps the authors could comment on this.

Reviewer #2:

In this report, Carlson and co-workers combine SILAC labeling, reconstitution of crude membranes using peptidiscs, size fractionation and mass-spec to characterize the interactome of the membrane proteins of *E. coli*. This is an ambitious study of potential interest that may contribute to future studies of PPI interactions of the membrane proteome. However, there are some serious problems with this study that in my opinion make it unsuitable in its current state for publication in *eLife*.

From a technological perspective, the novelty is incremental: the methodologies have been used in the past, including the peptidiscs. If I understand correctly, the only novelty is the use of the peptidiscs instead of detergent or amphiphilic copolymers.

From a biological perspective, all of the interactions that were used for validation have been previously described. The paper presents no new novel information.

Examination of Supplementary file 3 reveals that the interactions are heavily biased towards some proteins that are highly represented, while others are completely missing. In addition, many well-characterized interactions are not detected, while other interactions, that seem non-specific false positives are "detected" (see for examples the interactions of NADH dehydrogenase, BtuB, BamC, LptB, CydC).

In addition, below are some specific concerns that question the validity of the findings:

>The FepBCD complex is larger than MetINQ, yet the latter elutes at an earlier time. Along the same lines, the ATP synthase, which is considerably larger than both ABC transporters also elutes at a similar volume, as do BamABCD and NuoABCDEFGI. This is very suspicious. Can we be looking at aggregates? What is the void volume of this column?

The suggestion that MetQ is anchored to the membrane is new and unexpected. It would greatly add to the paper if this suggestion is experimentally validated with the proper controls of an SBP that is not expected to be membrane anchored.

In addition, why does this explain the co-elution with MetIN? This only suggests membrane association, and a stable transporter-SBP association.

The methodology requires detergent extraction, unlike the use of amphiphilic copolymers. This possibly disturbs many of the endogenous interactions. This issue is not discussed.

"The end result of reconstitution is a membrane protein that displays increased stability": unsupported statement. How many cases were examined?

From the width of the elution peaks of the SEC fractionation it appears that many of the interactions (the FPs) may stem from similar migrations rather than associations. This also likely explains the non-specific interactions that were detected.

For removal of DDM, the authors use dilution and re-concentration in a centricon. This is a major flaw since DDM also gets concentrated in this process (this phenomenon is easily detected using sugar detection methods). So, the assumption that the preparation contains no (or little) detergent is probably wrong.

"Upon removal of aggregate": At what stage? what does this mean?

How is the data MsbA data relevant? This study is supposed to provide proteomic information, so how is the behavior of a single protein relevant? One can relate to the other many advantages of DDM, but the behavior of MsbA is irrelevant. If the authors had chosen EmrE as their benchmark, would they extract the membranes using chloroform?

"As expected, a large fraction of these proteins (591 proteins) are predicted, based on gene ontology terms, to be associated with the cell envelope". Considering that this is a membrane preparation, this is not a large fraction. The membrane proteome is supposed to be 1/3 of all proteins so this is a pretty mild enrichment factor. This is actually quite surprising and should be addressed by the authors. Is there some common theme to the non-membrane proteins that were identified? Is there a common theme to the membrane proteins that are missing?

Subsection “Large membrane protein complexes are captured in the peptidisc library”: How many interactions were analyzed to support this statement?

On what basis the three analyzed cases (Figure 3) were chosen? Was this choice arbitrary?

The lack of rigorous analysis of any novel interaction is greatly missing.

The 13 Type-I SBPs that were identified by the authors: if these are not membrane anchored (like MetQ) and do not stably interact with their ABC transporters, how is it that they were identified? In this respect, why were other Type II SBPs (e.g., HmuT, FhuD, BtuF) not identified?

Reviewer #3:

The work by Carlson et al. describes a derivative of protein-correlation-profiling (PCP) for membrane complexomes based on stabilization of detergent-extracted membrane protein complexes by peptidiscs. Overall the paper is well written and good to follow and the experiments suited the needs.

It is clearly laid out that better approaches are needed for proteomics-based identification of protein-protein interactions of membrane protein complexes. The downsides of current technologies are discussed well. The approach presented here may be an improvement over other existing methods but the based on the data presented, the improvement does not seem to be impressive. Saying this, it is surprising to see that the authors apparently claim that a precision of 50% is sufficient to call something an identified interaction. It is not surprising to see then, that only a small proportion of these 50% overlap with previous analyses and that the validating experiments validate only very few of the found interactions. Interestingly, the authors set a threshold of 80% and 75% precision, respectively, for the validation experiments. First of all, this is inconsistent within the validation experiments (75% vs. 80%) and secondly, this is inconsistent with the previously set threshold of 50%, which the authors don't seem to trust themselves. It would be very beneficial for the paper if a false discovery rate could be calculated and applied because the question clearly remains, which of the identified interactions are trustworthy and which ones not.

Along the same lines, it would be good to show where the 340 overlapping interactions between this study and the one by Babu et al., 2017 rank on the graph in Figure 4C.

In the validation experiments using MipA, the authors identify YajC and AtpF as potential interacting proteins. The authors should discuss whether this makes sense in the light of the membrane and complex topology of these proteins.

A major current problem is the stabilization and identification of membrane protein complexes that bridge inner and outer membranes of Gram negative bacteria. The paper would benefit from presenting and discussion the subcellular localization of the identified complexes.

[Editors’ note: what now follows is the decision letter after the authors submitted for further consideration.]

Thank you for resubmitting your work entitled "Profiling the *E. coli* membrane interactome captured in Peptidisc libraries" for further consideration at *eLife*. Your revised article has been favorably evaluated by Richard Aldrich as the Senior Editor, a Reviewing Editor, and three reviewers.

The manuscript has been improved but there are some remaining issues that need to be addressed before acceptance, as outlined below:

Summary of paper:

The authors report on the application of their recently developed peptidisc approach to identify protein-protein interaction of the *E. coli* membrane proteome. Although other methods are available, the authors make the argument that the peptidisc approach holds advantages since it uses a membrane mimetic that is broadly applicable to membrane proteins of various sizes and to large complexes. In addition, the peptidisc scaffold protein also traps endogenous lipids that may be important for protein interactions and function. The authors benchmark their findings against interactome analysis provided via alternative approaches and demonstrate that their method is comparable, if not superior, especially for multi-component large complexes. The peptidisc interaction analysis validates known interactions but also identifies an unexpected stable interaction between the methionine substrate binding protein (MetQ) and its cognate ABC transporter (MetNI). The findings and the approach lay the foundations for future work examining the newly-identified interactions and the application of the approach to membrane-interactome analysis of other organisms.

Opinion:

The problem of identifying novel membrane protein interactions is important and interesting, and the development of new and simpler approaches is pertinent. However, in its present state, even after the previous round of revisions, it is unclear whether the peptidisc approach presents sufficient novelty to justify publication in *eLife*. Major revisions are needed to address the outstanding issues listed below.

Major issues:

1) The main claim of the authors is that their approach is a useful tool to identify novel interactions of the membrane proteome. However, of the newly-identified interactions they validate a single one (MetQ-MetIN), which is also not entirely surprising based on two recent publications from the Rees lab. It would greatly benefit the paper, and convince the reader of the usefulness of the approach, if the authors show a few more examples that demonstrate novel and unpredicted interactions.

2) The authors use 0.8-1% detergent (DDM) for 1 hour, yet claim this is a short time. However, most membrane proteins are fully extracted in a shorter time. Along the same lines, the authors show that the SEC and profiles obtained in DDM and in peptidiscs are similar. Since the use of detergent alone is simpler than that of the peptidiscs, it remains unclear how the presented approach is advantageous. To support this claim, the authors need to show a complete and rigorous comparison between DDM and peptidiscs. How many of the interactions identified by the peptidiscs approach would be lost if detergent only is used? The authors are encouraged to equally minimize the detergent exposure time in these experiments.

3) The authors use a detergent (DDM) concentration that at the conditions used (temperature, salt concentration) is likely above the CMC. As a result, the detergent is probably concentrated along with the peptidisc library. The authors need to measure the DDM concentration during, and at the end of, the concentration step to convince the reader that their approach is indeed superior to the use of detergent alone.

4) The authors prepare membrane fractions using ultracentrifugation, rather than density gradients or floatation assays. This leads to sedimentation of large proteins, aggregates, and complexes that are unrelated their analysis and confound the interpretation of the data. The authors must explain why they have chosen this approach and demonstrate that it does not undermine their analysis.

5) Several of the concerns raised by the reviewers during the initial round of submission remain inadequately answered. For example, the identification of 13 Type-I SBPs remains unexplained. The authors need to directly and clearly address this and other concerns raised by the reviewers.

6) The paper suffers from overstatement and inaccurate citations.

7) The supplementary tables should be reformatted. Now they are unintelligible.

---

## [Author Response]

[Editors’ note: the author responses to the first round of peer review follow.]

Reviewer #1:The manuscript describes a new approach for extracting and maintaining interactions in peptidiscs. This a very interesting paper worthy of publication in eLife. The study is complementary, yet distinct from nanodiscs and SMALPS, and appears to have some unique advantages.

We thank the reviewer for highlighting the novelty of our study. We have made substantial revisions to improve the quality of the manuscript.

Although the membrane proteome is still extracted with detergent prior to reconstitution results are similar to those obtained in SMALPs and nanodiscs. The number of unique insights obtained through the combination of methods applied here was however difficult to tease out.

There are important differences in the results we obtain in our side-by-side comparison between the peptidisc and SMALPs libraries. Although the solubilization efficiencies of the peptidisc and SMALPs are comparable (see Figure 3A and 3B), we observe dramatic differences in terms of the stability of multi-subunit membrane complexes. We compared the coelution profiles for three well-known complexes – the Bam complex, the respiratory chain complex and the ATP synthase complex – in both SMALPs and peptidiscs (see Figure 3C and 3D; Figure 3—figure supplement 1). In every case, the complexes are preserved in the peptidisc but largely dissociated in SMALPs. We have clarified this important point in the manuscript.

The nanodisc was not used in this report. We have, however, recently shown that – unlike the nanodisc – the peptidisc is a universal, “one-size-fits-all” reconstitution scaffold, able to reconstitute membrane proteins of differing sizes and topologies without needing addition of exogenous lipids (Carlson et al., 2018). In addition to streamlining the library reconstitution process, use of the peptidisc rather than the nanodisc also mitigates possible bias during the reconstitution process based on protein size. We have commented on these points in our revised Discussion.

It seems that the most important one was that MetQ co-elutes as a complex with MetNI. MetQ appears to be the only substrate binding protein that is lipid modified at its N-terminus. This I thought particularly interesting.

We agree with the reviewer. The finding that MetQ is lipidated and that it co-elutes with MetNI is an important and novel discovery. In this resubmission, we incorporate new data to rigorously validate these observations. Specifically: i) we validate the stability of the MetNIMetQ interaction by SILAC AP-MS (now included as Figure 8) and ii) we demonstrate the importance of MetQ lipidation by demonstrating that a non-lipidated MetQ mutant does not interact with MetNI (now included as Figure 8—figure supplement 2). We are also including an additional novel observation which we missed in our first submission. Our pairwise interaction list identifies NlpA (Lipoprotein 28) as a potential interactor of MetQ. We found this particularly interesting in light of a previous report which showed through complementation assays that NlpA may be able to deliver Methionine to the MetNI transporter in a MetQ-deleted strain, albeit at low efficiency (Zhang et al., 2003). Notably, a physical interaction between MetNI-Q and NlpA has not been reported to date in the literature. Our new MetNI SILAC AP-MS data now reveal significant enrichment of NlpA along with MetQ (Figure 8). This observation, along with our PCP-SILAC data, provide strong evidence that a physical interaction between MetNI-Q and NlpA does exist in the cell. Altogether, this further validate the peptidisc in untargeted interactomics studies.

Of the SecYEG translocon interacting partners it as surprising to me that simultaneous over- production of the SecYEG core translocon with both PpiD and YfgM is necessary for detection of this interaction. It might suggest that original interactions which are thought to be multiple are disrupted by the initial detergent solubilisation step. Perhaps the authors could comment on this.

We agree with the reviewer. We are not discounting the dissociative effects of detergent on the YfgM-PpiD-SecYEG interaction. We note that Digitonin, rather than DDM, was employed in the original paper which identified this complex (see (Götzke et al., 2014). It is therefore possible that an interaction between YfgM-SecYEG and/or PpiD-SecYEG would be preserved in Digitonin but dissociated in DDM (as done in our study). However, it is equally possible that strong interaction largely depends on correct stoichiometric ratio between components as our data seem to indicate (Figure 6—figure supplement 1). We have modified the manuscript to include a comment on this point.

Reviewer #2:In this report, Carlson and co-workers combine SILAC labeling, reconstitution of crude membranes using peptidiscs, size fractionation and mass-spec to characterize the interactome of the membrane proteins of E. coli. This is an ambitious study of potential interest that may contribute to future studies of PPI interactions of the membrane proteome. However, there are some serious problems with this study that in my opinion make it unsuitable in its current state for publication in eLife.

We agree with the reviewer that our initial version had certain flaws and accordingly, we have substantially revised the manuscript including additional computational and experimental validations, as well clarification both in the text and our responses above and below. However, as the reviewer also points out, this study potentially represents a major technical advance in the field of membrane interactomics (as such we submitted it as a “Tools and Resources” article).

From a technological perspective, the novelty is incremental: the methodologies have been used in the past, including the peptidiscs. If I understand correctly, the only novelty is the use of the peptidiscs instead of detergent or amphiphilic copolymers.

The principle novelty of our study is the combination of the peptidisc – a novel membrane mimetic system our laboratory developed recently (see (Carlson et al., 2018)- with the PCP-SILAC workflow. We contend that our study provides a significant technological and methodological advance for the following key reasons:

i) The peptidisc solves a crucial problem in the design of detergent free protein libraries because it enables “flexible”, streamlined reconstitution of membrane proteins and complexes of varying topologies and sizes (see (Carlson et al., 2018). This is in marked contrast to the nanodisc system, which may bias reconstitution toward membrane proteins and complexes of a certain size, based on the length of the scaffold protein used.

ii) From our results with SMA as shown in Figure 3 and Figure 3—figure supplement 1, we can say that the peptidisc is a comparatively gentle technique that is better suited for examining native protein interactions. Many membrane protein complexes which are preserved in the peptidisc are dissociated by the SMA polymer. The reviewer is referred to our response to point 2 from reviewer #1 which addresses this topic in detail.

From a biological perspective, all of the interactions that were used for validation have been previously described. The paper presents no new novel information.

We are presenting this study without emphasis on discovery of novel interactions, but rather as a way to rapidly assess a cell membrane interactome. Nevertheless – as stated in our response to Point 3 from reviewer #1 – we are including new data with this resubmission which reveal novel insights into the ABC transporter MetNI-MetQ. Specifically, we reveal that MetQ is a unique SBP in *E. coli*, because it is tethered to the membrane by an N-terminal lipid anchor. We further show that this lipidation is required for stable association between MetQ and MetNI. Furthermore, we identify an interaction and validate NlpA (Lipoprotein 28) as an interactor of the MetNI-Q complex (data presented in Figure 8 and Supplementary Table 1G). These are novel discoveries with profound implications for our understanding of the mechanism of MetNI-Q mediated Methionine import in bacteria. These data are nicely validating our peptidisc interactomic approach.

Examination of Supplementary file 3 reveals that the interactions are heavily biased towards some proteins that are highly represented, while others are completely missing.

We agree with the reviewer that our interactome contains highly represented proteins. This likely reflects, at least in part, the true biological promiscuity of some proteins. Proteins participating in medium-to-large complexes can have dozens of interacting partners. For example, the most represented protein in our interactome, P0AEH1 (YaeL), which we detect in 114 interactions, is listed as occurring in 106 interactions in the most recent BioGRID database (https://thebiogrid.org/4261736).

In our interactome, the proportion of very highly represented proteins, *e.g*. the number of proteins in >100 interactions as a fraction of all unique proteins in the interactome, is comparable to other interactomes: we find 1.6% of unique proteins in >100 interactions (9/562), compared to 0.4%, 1%, 0.2%, and 3% (Babu et al., 2018; Hein et al., 2015; Wan et al., 2015; Scott et al., 2017). While some interactions for highly represented proteins will be false positives, we aimed to control for the false positive rate with multiple computational validations. Regarding missing interactions and controlling for false positives, please see our response just below.

In addition, many well-characterized interactions are not detected, while other interactions, that seem non-specific false positives are "detected" (see for examples the interactions of NADH dehydrogenase, BtuB, BamC, LptB, CydC).

We agree with the reviewer that, ideally, we would capture all well-characterized interactions. Unfortunately, this is beyond the current state of interactome studies. For example, taking the CORUM database as the “full” human interactome of >80,000 interactions, well-cited interactome studies (Hein et al., 2015) and (Wan et al., 2015) capture only 3% and 7% of pairwise CORUM interactions, respectively. Similarly for *E. coli*, (Babu et al., 2018) captures 40 of the 993 known protein interactions we use as reference in our study.

Some proteins (hence interactions) are missing because they are low abundance or otherwise hard to detect with mass spectrometry (e.g. poor ionization). Importantly, though, many of the interactions we do detect are indeed well-characterized interactions, such as the ATP synthase complex, the Bam complex, and the respiratory chain complex (see Figure 3 and

Figure 3—figure supplement 1).

Because it is difficult to capture known interactions, a common strategy of interactome studies (and one we take here) is to accept non-specific false positives as the cost of detecting true interactors. It is therefore very important to estimate and report the proportion of false positives. We took pains to do that here by *i*) calculating the TP-to-FP ratio as measured against our reference *E. coli* complexes; *ii*) calculating how many interactions share GO terms, binding domains, and have correlated expression levels, which are measures of true, biologically interacting proteins; and *iii*) cross-validating our detected interactions against independent datasets, namely the cell envelope (CE) interactome (Babu et al., 2018) and the two validating interactomes.

In addition, below are some specific concerns that question the validity of the findings:The FepBCD complex is larger than MetINQ, yet the latter elutes at an earlier time. Along the same lines, the ATP synthase, which is considerably larger than both ABC transporters also elutes at a similar volume, as do BamABCD and NuoABCDEFGI. This is very suspicious. Can we be looking at aggregates? What is the void volume of this column?

Size exclusion chromatography was performed using two BioSep4000 columns (Phenomenex) connected in tandem. The total volume of the column was 18 mL and the void volume was 6 mL. The void volume is represented as the zero on the x-axis of all co-elution graphs in the manuscript. Thus, any proteins eluting past the zero point in our dataset are fully soluble. We have clarified this last point in legend of Figure 3 as it appears first.

We were also puzzled by the elution behaviour of some protein complexes on the silica resin. Our main hypothesis to explain this behaviour is that certain membrane protein complexes may retain significant amounts of phospholipids, which increase the apparent molecular weight of the complex. These additional lipids may also affect the chromatographic behaviour of the proteins during size exclusion chromatography on silica – specifically, by broadening elution peaks and thus decreasing the resolution of the column. Given the broadness of the peaks we observed, we were concerned about detecting “false positive” interactions caused by co-elution of noninteracting proteins. To filter out false positive interactions caused by spurious co-elution, we incorporated information from a co-expression dataset (M3D) (see Faith et al., 2008).

The suggestion that MetQ is anchored to the membrane is new and unexpected. It would greatly add to the paper if this suggestion is experimentally validated with the proper controls of an SBP that is not expected to be membrane anchored.

Reviewer #1 also asked for confirmation and validation of this important finding. Please see our response to Point 3 from reviewer #1 above. Briefly, we co-expressed MetNI with MetQ and validated the stability of the interaction by showing that MetQ co-purifies with his-tagged MetNI. In this experiment, we noticed that MetQ is localized to the membrane fraction, an observation which supports our suggestion that MetQ is lipidated.

To explore the importance of MetQ lipidation for interaction with MetNI, we constructed a mutant – MetQ C23A – in which the Cysteine residue required for lipidation is replaced with an Alanine. Although this mutant is localized to the membrane fraction, it does not interact with MetNI, thus demonstrating the importance of MetQ lipidation for formation of the MetNI-Q complex. This series of experiments reveals the ability of the peptidisc-PCP-SILAC workflow for capturing and identifying novel, unexplored aspects of membrane protein complexes that are not evident using other experimental methods.

In addition, why does this explain the co-elution with MetIN? This only suggests membrane association, and a stable transporter-SBP association.

Our experiments using the non-lipidated MetQ C23A mutant reveal the importance of MetQ lipidation for its interaction with MetNI – please see the point above.

The methodology requires detergent extraction, unlike the use of amphiphilic copolymers. This possibly disturbs many of the endogenous interactions. This issue is not discussed.

We use minimal non-denaturing detergent, and importantly we rapidly remove it by replacing it with peptidisc, so that dissociation of complexes caused by detergent exposure is meant to be minimized. However, it is true that some complexes will be missed or destroyed. We are explaining this unavoidable caveat in the Discussion of the manuscript. A possible way around to this problem would to construct peptidisc libraries using three different detergents with partially overlapping solubilization solubilities (as has been done in Babu et al., 2018).

Regarding the amphipatic SMA co-polymer, we present data that rather indicate that SMA is not a “soft” solubilization agent. As shown Figure 3, some large complexes which can be isolated in the peptidisc are instead bracked apart when the membrane is solubilized with the amphiphilic SMA polymer. This drawback clearly limits the applicability of SMA as a solubilization agent for interactomics studies.

"The end result of reconstitution is a membrane protein that displays increased stability": unsupported statement. How many cases were examined?

This statement is supported by our previous publication, as well as by unpublished results from our laboratory (see (Carlson et al., 2018), in which we show a major increase in membrane protein stability upon reconstitution in peptidisc.

From the width of the elution peaks of the SEC fractionation it appears that many of the interactions (the FPs) may stem from similar migrations rather than associations. This also likely explains the non-specific interactions that were detected.

This is a caveat inherent to any PCP technique, and we agree that the elution peaks in this study are wider than we would have liked. However, the consequences of the wide peaks are testable: if they’re too wide, then true interacting proteins will be swamped out by spurious co-fractionating proteins, and we will be unable to predict true positives better than false positives. We did not find that this was the case. That is, we were able to predict many interactions that, through computational validation, are of comparable quality to other interactomes. Additionally to mitigate this effect, we use two columns arranged in tandem and we incorporate expression data to avoid spurious interaction identification. That is, interactions are predicted using our peptidisc co-fractionation data and the M3D expression database. In this resubmission, we have also incorporated interaction data from AP-MS studies to provide a high-confidence list of interactions. Altogether, while we agree that the wide elution peaks are in issue, we believe we’ve sufficiently accounted for them, given that our interactome looks relatively “high quality” (computational validation compares favourably to other published interactomes, interactions validated in by wet lab experiments, etc.).

For removal of DDM, the authors use dilution and re-concentration in a centricon. This is a major flaw since DDM also gets concentrated in this process (this phenomenon is easily detected using sugar detection methods). So, the assumption that the preparation contains no (or little) detergent is probably wrong.

The detergent is diluted below its CMC, so that monomeric detergent molecules can pass through the centricon filter. We cannot exclude the possibility that small remnant, yet non-solubilizing amounts of detergent are trapped in the peptidiscs.

"Upon removal of aggregate": At what stage? what does this mean?

Removal of aggregated material occurs during ultracentrifugation immediately following detergent solubilization of the membrane. We have clarified this in the manuscript.

How is the data MsbA data relevant? This study is supposed to provide proteomic information, so how is the behavior of a single protein relevant? One can relate to the other many advantages of DDM, but the behavior of MsbA is irrelevant. If the authors had chosen EmrE as their benchmark, would they extract the membranes using chloroform?

The MsbA data is important because it shows that our peptidisc library method traps individual membrane proteins and complexes into discrete peptidisc particles, rather than non-specifically clustering them together. To show this, we overproduced a model protein (i.e. MsbA) and subsequently purified it from the library. From this, we observed efficient isolation of MsbA homo-dimer – free from contaminants – and perfectly soluble, as evaluated on SDS-PAGE and CN-PAGE respectively. This data confirms that proteins are not randomly captured into peptidiscs. We have clarified this important point in the manuscript.

> "As expected, a large fraction of these proteins (591 proteins) are predicted, based on gene ontology terms, to be associated with the cell envelope". Considering that this is a membrane preparation, this is not a large fraction. The membrane proteome is supposed to be 1/3 of all proteins so this is a pretty mild enrichment factor. This is actually quite surprising and should be addressed by the authors. Is there some common theme to the non-membrane proteins that were identified? Is there a common theme to the membrane proteins that are missing?

The peptidisc libraries employed in this study were prepared from crude cell membranes. The only enrichment performed to prepare the membranes was an ultracentrifugation step following cell lysis. While ultracentrifugation leads to sedimentation of cell membranes, it is well documented that large, soluble cellular complexes – such as GroEL and ribosomes – also pellet along membranes (Papanastasiou et al., 2013, 2015). We have clarified this point in the manuscript.

Furthermore, it is well known that membrane proteins suffer from radically decreased identification rates in mass spectrometry experiments, which would further decrease the number of identified membrane proteins. The common theme among the non-membrane proteins we identify were large, soluble complexes that co-sediment with the membrane fraction (such GroEL, ribosome, AceP, etc.).

Subsection “Large membrane protein complexes are captured in the peptidisc library”: How many interactions were analyzed to support this statement?

We looked at all pairwise comparisons between complex members of the three complexes (ATP synthase, Bam, and respiratory complex). This gave 48 pairwise comparisons in the SMALP condition (21 + 6 + 21 comparisons, respectively) and 55 comparisons in the peptidisc condition (28 + 6 + 21 comparisons, respectively).

On what basis the three analyzed cases (Figure 3) were chosen? Was this choice arbitrary?

The validation targets were picked in large part due to their relevance with existing projects in our laboratory (Sec and ABC). The proteins YfgM and PpiD are relatively uncharacterized ancillary subunits of the bacterial Sec translocon, which is a major focus of study in our lab. As a result, many of the materials were already available to conduct these experiments (i.e. a stable SecYEG expression construct). The complex MetNI was chosen as a validation target due to the surprising observation that it co-elutes with the SBP MetQ (see above points for details). We included data on the scaffold protein MipA in our earlier submission. We now feel that inclusion of this data detracts from the main focus of our study, and have therefore removed it in this resubmission.

> The lack of rigorous analysis of any novel interaction is greatly missing.

Since our first submission, we have validated the MetNI-Q interaction, and we have demonstrated the importance of MetQ lipidation in this complex. This is an important and novel biological observation. Additionally, our newly included SILAC AP-MS experiment reveals the lipoprotein NlpA as an interactor of MetNI-Q. The MetNI-Q-NlpA association has only been characterized to a very limited extent in the current literature. We have now validated this novel interaction using AP-MS, which further validate the validity of the peptidisc-PCP approach.

However, we would like to stress out that our major aim was to demonstrate that our method is suitable for predicting interactomes from membrane proteins. Also, PCP-SILAC, while useful for identifying novel interactions, is a powerful tool for comparative interactomics. While we did not expand this study to a third label, the work still remains a valid proof of concept for subsequent analysis based in detergent-free fractionation of a membrane preparations.

The 13 Type-I SBPs that were identified by the authors: if these are not membrane anchored (like MetQ) and do not stably interact with their ABC transporters, how is it that they were identified? In this respect, why were other Type II SBPs (e.g., HmuT, FhuD, BtuF) not identified?

Type I SBPs have weak affinity for their cognate transporters, which may result in their co-sedimentation with the membrane fraction during ultracentrifugation following cell lysis. However, as reported by other biochemists including ourselves using the maltose transporter, this affinity is too weak to resist the dissociative effects of detergent during membrane solubilization. This interaction is also not restored until the transporter in stabilized in a very specific conformational state requiring non-hydrolyzable ATP analogs. This likely explains why most soluble Type 1 SBPs are eluted at the end of the fractionations and are fully dissociated from their cognate transporters.

Reviewer #3:The work by Carlson et al. describes a derivative of protein-correlation-profiling (PCP) for membrane complexomes based on stabilization of detergent-extracted membrane protein complexes by peptidiscs. Overall the paper is well written and good to follow and the experiments suited the needs.It is clearly laid out that better approaches are needed for proteomics-based identification of protein-protein interactions of membrane protein complexes. The downsides of current technologies are discussed well. The approach presented here may be an improvement over other existing methods but the based on the data presented, the improvement does not seem to be impressive. Saying this, it is surprising to see that the authors apparently claim that a precision of 50% is sufficient to call something an identified interaction. It is not surprising to see then, that only a small proportion of these 50% overlap with previous analyses and that the validating experiments validate only very few of the found interactions.

We agree with the reviewer that high level of false positives is an unwanted feature of interactomes. However, as explained above in response to reviewer #2, we view false positives as the cost of detecting true interactors in a high-throughput study. We believe the number of false positives needs to be estimated and controlled for, and we took care to do both of these. Further, one of the major aims in this study was to provide a larger list that researchers can reduce as needed if a higher confidence set of interactions is required. We provide two ways for researchers to refine the full published list (by interaction score and the “high confidence” subset) and we have revised our Discussion to better explain this to the reader less familiar with global interactome analysis. We believe this is the preferred method since it gives researchers more choice. That is, it is better to publish lower-scoring interactions and give researchers the option to use them or not, as opposed to not publishing them at all.

We agree that 50% precision is low, but we note that it is in the same range as other interactome studies. Defining true and false positives as we do in our study, and using the CORUM database as the true interactome, other well-cited studies (Hein et al., 2015) and (Wan et al., 2015) have precisions of 31% and 65%, respectively. Using our *E. coli* reference complexes as the true interactome, (Babu et al., 2018) has a precision of 32%. Therefore, we note that the proportion of false positives in our study is not unusually high compared to other interactome studies.

We agree that the overlap between our interactome and other interactomes is small in magnitude. However, we believe this does not invalidate our interactome for two reasons. First, because the full interactome of a species is estimated to be large (e.g. hundreds of thousands of interactions in the human interactome (Stumpf et al., 2008) and interactome studies sample a small, random subset of it, overlap between interactome studies tends to be small. For example, two well-cited, high-throughput human interactome studies with 31,000 and 14,000 interactions, respectively, only have 408 interactions in common (Wan et al., 2015; Hein et al., 2015), a much smaller relative overlap than our interactome and (Babu et al., 2018). Second, although small, the overlap between our interactome and other *E. coli* interactomes is significantly larger than expected by random chance (Figure 5D, Figure 5—figure supplement 1). Therefore, we believe the overlap between our interactome and the *E. coli* interactome in (Babu et al., 2018) is meaningful and supports the validity of our conclusions.

Interestingly, the authors set a threshold of 80% and 75% precision, respectively, for the validation experiments. First of all, this is inconsistent within the validation experiments (75% vs. 80%) and secondly, this is inconsistent with the previously set threshold of 50%, which the authors don't seem to trust themselves.

The reviewer raises a good point that, by design, using a threshold of 50% estimated precision produces an interactome with a large number of false positives. We chose this threshold because we wanted to provide a comprehensive list with fewer false negatives, at the at the cost of more false positives. Importantly and as mentioned above, we now explicitly say this in the Discussion to inform the reader. Further, we trust these interactions in proportion to their estimated precision level. Therefore we wanted to validate interactions with a higher estimated precision and to provide interpretable graphs, we used a higher cut-off for the validation experiments. Finally, from the immediately previous comment, 50% precision is in the range of well-cited interactome papers (31%, 32%, and 65% precision in (Wan et al., 2015; Babu et al., 2018; Hein et al., 2015), respectively).

It would be very beneficial for the paper if a false discovery rate could be calculated and applied because the question clearly remains, which of the identified interactions are trustworthy and which ones not.

The estimated false discovery rate for the peptidisc interactome can be calculated from the precision (FDR = 100% – precision). We agree this is an important point, and to make this connection explicit we now include the FDR calculation in the text.

Along the same lines, it would be good to show where the 340 overlapping interactions between this study and the one by Babu et al., 2017 rank on the graph in Figure 4C.

Interactions that overlap between our study and the interactome published in Babu et al., 2018 tend to be higher scoring than non-overlapping interactions. We added this information to Figure 4C, and we now discuss this in the text.

In the validation experiments using MipA, the authors identify YajC and AtpF as potential interacting proteins. The authors should discuss whether this makes sense in the light of the membrane and complex topology of these proteins.

MipA is a lipoprotein anchored on the periplasmic side of the outer membrane. Both YajC (part of the Sec complex) and AtpF (part of the F1F0 ATPase) are inner membrane proteins with exposed loops in the periplasm. Therefore, these interactions do make sense in light of the topologies of these proteins. Please note that – as stated above in response to reviewer #2 – we have removed the MipA validation experiment in this resubmission. In light of the additional in-depth validation data we are presenting on the MetNI-Q complex, we feel that inclusion of the MipA data detracts from the main focus of the manuscript.

A major current problem is the stabilization and identification of membrane protein complexes that bridge inner and outer membranes of Gram negative bacteria. The paper would benefit from presenting and discussion the subcellular localization of the identified complexes.

We identify a number of interactions that bridge between the inner and outer membranes. We observe interactions between the lipoprotein Pal (outer membrane), the periplasmic protein TolB and the inner membrane proteins TolQ and TolR. More in-depth characterization of these interactions is currently underway in our laboratory. As suggested by reviewer #1, we have amended Figure 1 so that the reader readily understand our study deals with a complex network of interactions taking place across a double cell membrane.

[Editors' note: the author responses to the re-review follow.]

[…] The problem of identifying novel membrane protein interactions is important and interesting, and the development of new and simpler approaches is pertinent. However, in its present state, even after the previous round of revisions, it is unclear whether the peptidisc approach presents sufficient novelty to justify publication in eLife. Major revisions are needed to address the outstanding issues listed below.

We thank the editor and reviewers for highlighting the importance of our study. We strongly feel that our work is sufficiently novel for publication in *eLife*, especiallygiven the scarcity of proteomic methods that effectively enable analysis of membrane protein interactions in aqueous solution. In this response, we are including additional new data to unequivocally demonstrate the superiority of the peptidisc approach relative to detergent for stabilizing and detecting novel interactions. We are showing the following:

i) We identify a novel interactor of the MetNI transporter, the lipoprotein NlpA. Using SILAC AP/MS, we demonstrate that this interaction is preserved in peptidisc and undetected in detergent (Figure 8).

ii) We discover that MetQ is a lipidated SBP and we demonstrate the importance of this membrane anchor for association to the MetNI transporter (Figure 8—figure supplement 2). The role of MetQ lipidation has been overlooked in earlier studies (1,2).

iii) We show that YfgM and PpiD are two membrane-embedded chaperones that interact strongly with the Sec translocon and that a proper stoichiometry is important for detection of this association (Figure 6—figure supplement 1). Similar to what we observe with MetNI-NlpA, these interactions are difficult to capture in detergent (Figure 8), which may explain why these interactions have been considered transient and detected only indirectly in the past (3,4).

iv) Continuing with the Sec translocon, our pairwise interaction dataset reveals an astonishing correlation with the Bam complex. We confirm this observation using SILAC AP/MS (Figure 6). There significant enrichment of the BamB and BamC subunits and here also, these interactions are not detected when the same workflow is performed in detergent. Further work will be necessary to understand the interface of association, but the identification of this trans-membrane super-complex has profound implications for our understanding of outer membrane protein biogenesis. In further support, while our manuscript was under revision, a study from the Collinson group was released in bioRvix.org. Their work reports the initial isolation and a low resolution EM structure of a Bam-Sec translocon (https://doi.org/10.1101/589077). This second independent study from a leader in the field of Sec-mediated protein translocation further underscores the importance the Bam-Sec interaction we have identified.

v) Continuing with the Bam complex, we show that all 5 subunits are fully captured in peptidisc in addition to two other interactors – RcsF and OmpA (Figure 7). These interactors were previously inferred from genetic and in vivo cross-linking experiments but never biochemically isolated (5,6). Accordingly, we show that the BamA-RcsF-OmpA interaction is less apparent in detergent than in peptidisc (Figure 7).

We further explain these findings in our point-by-point response below. We also provide a “compare manuscript” document to show the extend of the changes we have made.

Major issues:1) The main claim of the authors is that their approach is a useful tool to identify novel interactions of the membrane proteome. However, of the newly-identified interactions they validate a single one (MetQ-MetIN), which is also not entirely surprising based on two recent publications from the Rees lab. It would greatly benefit the paper, and convince the reader of the usefulness of the approach, if the authors show a few more examples that demonstrate novel and unpredicted interactions.

Our findings on MetNI-Q are novel because they reveal the importance of MetQ lipidation for association to the MetNI transporter. The role of MetQ lipidation for mediating this interaction has never been reported before, and this lipidation turns out to be a unique case among the Type I family of ABC transporters in *E. coli*. The recent papers from the Rees lab (Biol Chem 2015 and PNAS 2018) report on the interaction of MetQ-MetNI and high-resolution crystal structure of the complex. However, their work employs a truncated non-lipidated MetQ mutant, which was engineered to artificially increase its affinity to MetNI (1,2). Thus, our findings are novel and furthermore, they call for a re-evaluation of the role of MetQ lipid anchor for complex stability and substrate transport. In our Results, we present data showing the importance of MetQ lipidation. We co-express his-tagged MetNI with either wildtype MetQ or mutant MetQ-C23A with a disrupted lipobox. We find that only wild-type MetQ co-purifies with MetNI, thus demonstrating the importance of MetQ lipidation for the formation of the MetNI-Q complex.

As an additional response to the concern of novelty – and to highlight the advantage of the peptidisc over detergent – we report for the first time an interaction of the MetNI transporter with the lipoprotein NlpA (also termed Lipoprotein 28). This interaction was identified in our pairwise interaction dataset with a high degree of confidence (>80%). Using SILAC AP/MS, we provide direct evidence that MetNI is interacting with NlpA. Interestingly, the same experiment in detergent fails to detect this interaction. Therefore, we confirm using two different methods (AP/MS and PCP-SILAC) that our Peptidisc workflow is able to reveal novel protein interactions. Interestingly, an earlier study back in 2003 has suggested that NlpA can deliver methionine to the MetNI transporter in a MetQ-deleted strain (7). This biological finding supports the relevance of the physical interaction between NlpA and MetNI.

To provide additional examples that interactions are preserved in the peptidisc, we employ the Sec translocon. Our peptidisc interaction list reveal a number of known and unknown interactors of the Sec translocon at high degree of precision (>75%). These include the membrane-bound chaperones YfgM and PpiD. To validate these interactions, we express the SecY complex in SILAC conditions and performed AP/MS in both detergent and peptidisc. The interactions between the SecY complex and the membrane-bound YfgM and PpiD are well preserved in peptidisc but are largely disrupted in detergent.

Our peptidisc data list also reveals surprising correlation between the Sec translocon subunits and the outer membrane Bam complex. Interactions between SecYEG and the BamA, BamC and BamD subunits in particular are present in our interaction list at high precision. Our SILAC AP/MS data now provides direct supporting additional evidence for this remarkable interaction. There is also strong enrichment of the porin OmpA along BamB and BamC subunits. As above with YfgM and PpiD, the interaction between the Sec translocon and the Bam complex is not evident when the same experiments are performed in detergent.

The BamABCDE complex is also employed as a third validation target. We expressed BamA in SILAC labeling conditions and performed AP/MS in both detergent and peptidisc. As expected, we observe enrichment of all subunits of the Bam complex in peptidisc, which is not the case in detergent. We also observe enrichment of the inner membrane proteins YidC and YajC, both of which are ancillary subunits of the Sec translocon. This later result corroborates our earlier AP/MS results with the Sec translocon.

Analysis of the Bam pairwise interaction dataset indicates two other interactors, RcsF and OmpA. An interaction of the Bam complex-RcsF has been previously reported using indirect methods such as genetic complementation and in vivo photo-crosslinking, but has not been biochemically isolated (5,6). Consistent with these previous studies, both OmpA and RcsF are significantly enriched along with BamA in our peptidisc SILAC AP/MS. When the same BamA pulldown experiment is performed in detergent, only the BamC and BamD subunits appear enriched – suggesting that the Bam complex is somewhat labile in detergent as previously reported (8). Additionally, the only other interactor that is significantly enriched is RcsF but not OmpA. These observations highlight again the advantages of the peptidisc method for preserving and identifying transient interactions.

2) The authors use 0.8-1% detergent (DDM) for 1 hour, yet claim this is a short time. However, most membrane proteins are fully extracted in a shorter time. Along the same lines, the authors show that the SEC and profiles obtained in DDM and in peptidiscs are similar. Since the use of detergent alone is simpler than that of the peptidiscs, it remains unclear how the presented approach is advantageous.

The time exposure (0.8-1% DDM for 1 hour) correspond to the initial solubilization step, which is performed at high protein and lipid content. The subsequent fractionation step is performed without detergent. This treatment is therefore short compared to the time exposure that occurs during fractionation in detergent conditions (i.e. several hours with continuous input of new micelles during elution, thereby increasing protein delipidation and subunits dissociation). We have revised the manuscript to clarify this point.

We show in Figure 2 that the protein profiles in detergent DDM and in peptidisc are visually similar, as assessed by SDS-PAGE and Coomassie staining. However, mass spectrometry analysis of the detergent and peptidisc samples reveals significant differences, that is 125 IDs in detergent versus 162 IDs in peptidisc for the most protein-rich fraction. This lesser identification number in detergent is likely due to the acetone precipitation step that is required before MS analysis. Since peptidisc does not require removal of detergent before MS, the acetone step in omitted, and the method yield higher protein ID number which is especially important when dealing with low abundance or hard to detect membrane proteins.

To support this claim, the authors need to show a complete and rigorous comparison between DDM and peptidiscs. How many of the interactions identified by the peptidiscs approach would be lost if detergent only is used? The authors are encouraged to equally minimize the detergent exposure time in these experiments.

As mentioned above, we have included side-by-side comparisons to support our claim that the peptidisc is superior to detergent. Specifically, we have included SILAC-based AP/MS experiments to compare in a quantitative manner the detergent and peptidisc methods. From these additional datasets, it is evident that the peptidisc preserves interactions that are dissociated or simply lost due to the detergent treatment.

3) The authors use a detergent (DDM) concentration that at the conditions used (temperature, salt concentration) is likely above the CMC. As a result, the detergent is probably concentrated along with the peptidisc library. The authors need to measure the DDM concentration during, and at the end of, the concentration step to convince the reader that their approach is indeed superior to the use of detergent alone.

The peptidisc library is water-soluble since its membrane protein content can be fractionated by size-exclusion chromatography in the complete absence of detergent (Figure 2). We also show that the peptidisc library can be visualized on native-PAGE, in contrast to the detergent sample that forms large aggregate at the top of the gel during migration (Figure 2E, compare lane 1 to lane 2). This experimental evidence allows to conclude that detergent has been removed and replaced by the peptidisc during the reconstitution step using filtration. Evidently, the pore of the filtration device is large enough to allow detergent monomers to pass through. The dynamic behavior of detergent micelles has well known and the auto-assembly process that occur upon removal of detergent during filtration is well documented with the nanodisc.

4) The authors prepare membrane fractions using ultracentrifugation, rather than density gradients or floatation assays. This leads to sedimentation of large proteins, aggregates, and complexes that are unrelated their analysis and confound the interpretation of the data. The authors must explain why they have chosen this approach and demonstrate that it does not undermine their analysis.

We agree, purification of the *E. coli* inner membrane traditionally employs fractionation using density gradient centrifugation after initial ultracentrifugation (3,9,10). These additional steps serve to remove outer membranes, peripherally associated proteins and large complexes such as ribosomes (10). We did not perform these additional fractionation steps because we wanted to cover and preserve as many cell envelope protein-protein interactions as possible; that is, we wanted to preserve peripheral and outer membrane proteins along with inner membrane proteins in our peptidisc libraries. We note that the peptidisc library preparations were fractionated by size exclusion chromatography to remove protein aggregates.

We did not find evidence that ribosomes undermine our analysis. In fact, large complexes such as the ribosomes are important markers for the downstream bioinformatic analysis, since these complexes are employed as gold standard reference. Our machine learning approach uses these gold standard references as training labels, and more training data provides a better-calibrated classifier and thereby improves our ability to predict interacting proteins. Filtering these nonmembrane complexes from the data would prevent their use as training data, thus reducing our ability to give confidence on our interacting membrane protein dataset.

5) Several of the concerns raised by the reviewers during the initial round of submission remain inadequately answered. For example, the identification of 13 Type-I SBPs remains unexplained. The authors need to directly and clearly address this and other concerns raised by the reviewers.

The abundant Type I SBPs have weak affinity for their cognate transporters, yet this affinity (and abundance) is sufficiently high to allow their co-sedimentation with the membrane fraction during initial ultracentrifugation. This explains their presence in our crude membranes stock which as mentioned above, are not purified by density gradient ultracentrifugation. The Type I SBP-transporter complex are unable to survive the downstream detergent solubilization and reconstitution steps, except when a lipid anchor exists – as is the case with MetQ. Thus, the vast majority of Type I SBPs are fully dissociated from their cognate transporters and therefore elute in the late fractions of the size exclusion chromatography.

6) The paper suffers from overstatement and inaccurate citations.

We have edited the manuscript to limit over-statements and we have asked expert co-authors on this manuscript to verify accuracy of the citations.

7) The supplementary tables should be reformatted. Now they are unintelligible.

This manuscript went thought two rounds of revisions which has led to inclusion of many additional tables. We have re-formatted some of these supplementary tables to improve clarity.

References:

1) Nguyen, P. T., Li, Q. W., Kadaba, N. S., Lai, J. Y., Yang, J. G., and Rees, D. C. (2015) The contribution of methionine to the stability of the *Escherichia coli* MetNIQ ABC transporter-substrate binding protein complex. Biol Chem 396, 1127-1134,

2) Nguyen, P. T., Lai, J. Y., Lee, A. T., Kaiser, J. T., and Rees, D. C. (2018) Noncanonical role for the binding protein in substrate uptake by the MetNI methionine ATP Binding Cassette (ABC) transporter. Proc Natl Acad Sci U S A 115, E10596-E10604.

3) Maddalo, G., Stenberg-Bruzell, F., Götzke, H., Toddo, S., Björkholm, P., Eriksson, H., Chovanec, P., Genevaux, P., Lehtiö, J., Ilag, L. L., and Daley, D. O. (2011) Systematic analysis of native membrane protein complexes in *Escherichia coli*. J Proteome Res 10, 1848-1859.

4) Götzke, H., Palombo, I., Muheim, C., Perrody, E., Genevaux, P., Kudva, R., Müller, M., and Daley, D. O. (2014) YfgM is an ancillary subunit of the SecYEG translocon in *Escherichia coli*. J Biol Chem 289, 19089-19097.

5) Hart, E. M., Gupta, M., Wühr, M., and Silhavy, T. J. (2019) The Synthetic Phenotype of Δ. MBio 10.

6) Konovalova, A., Perlman, D. H., Cowles, C. E., and Silhavy, T. J. (2014)

Transmembrane domain of surface-exposed outer membrane lipoprotein RcsF is threaded through the lumen of β-barrel proteins. Proc Natl Acad Sci U S A 111, E4350-4358.

7) Zhang, Z., Feige, J. N., Chang, A. B., Anderson, I. J., Brodianski, V. M., Vitreschak, A. G., Gelfand, M. S., and Saier, M. H. (2003) A transporter of *Escherichia coli* specific for L- and D-methionine is the prototype for a new family within the ABC superfamily. Arch Microbiol 180, 88-100,

8) Gu, Y., Li, H., Dong, H., Zeng, Y., Zhang, Z., Paterson, N. G., Stansfeld, P. J., Wang, Z., Zhang, Y., Wang, W., and Dong, C. (2016) Structural basis of outer membrane protein insertion by the BAM complex. Nature 531, 64-69,

9) Stenberg, F., Chovanec, P., Maslen, S. L., Robinson, C. V., Ilag, L. L., von Heijne, G., and Daley, D. O. (2005) Protein complexes of the *Escherichia coli* cell envelope. J Biol Chem 280, 34409-34419.

10) Papanastasiou, M., Orfanoudaki, G., Koukaki, M., Kountourakis, N., Sardis, M. F., Aivaliotis, M., Karamanou, S., and Economou, A. (2013) The *Escherichia coli* peripheral inner membrane proteome. Mol Cell Proteomics 12, 599-610,